# NEURAL CONCEPT VERIFIER: SCALING PROVER-VERIFIER GAMES VIA CONCEPT ENCODINGS

## ABSTRACT

While *Prover-Verifier Games* (PVGs) offer a promising and much needed path toward verifiability in nonlinear classification models, they have not yet been applied to complex inputs such as high-dimensional images. Conversely, *Concept Bottleneck Models* (CBMs) effectively translate such data into interpretable concepts but are limited by their reliance on low-capacity linear predictors. In this work, we push towards real-world verifiability by combining the strengths of both approaches. We introduce *Neural Concept Verifier (NCV)*, a unified framework combining PVGs for formal verifiability with concept encodings to handle complex, high-dimensional inputs in an interpretable way. NCV achieves this by utilizing recent minimally supervised concept discovery models to extract structured concept encodings from raw inputs. A *prover* then selects a subset of these encodings, which a *verifier*, implemented as a nonlinear predictor, uses exclusively for decision-making. Our evaluations show that NCV outperforms CBM and pixel-based PVG classifier baselines on high-dimensional, logically complex datasets and also helps mitigate shortcut behavior. Overall, we demonstrate NCV as a promising step toward concept-level, verifiability AI.

## 1 INTRODUCTION

Deep learning has achieved remarkable predictive performances, but often at the expense of *interpretability and trustworthiness* (Rudin, 2019). However, particularly in high-stakes applications, it is critical that models provide *verifiable justifications* for their decisions (Irving et al., 2018; Fok & Weld, 2023). *Prover-Verifier Games* (PVGs), introduced by Anil et al. (2021), formalize such justifications via a game-theoretic approach: a prover provides evidence to convince a verifier, who accepts only verifiable proofs. A prominent instantiation of PVGs is the *Merlin-Arthur Classifier* (Wäldchen et al., 2024), *i.e.*, a classifier guided by cooperative and adversarial provers, offering formal interpretability guarantees through information-

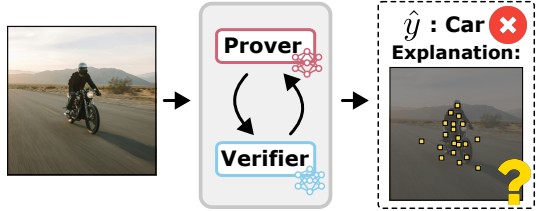

Figure 1: Challenges of Prover-Verifier Games (PVGs) in image classification: (i) It is non-trivial to scale up for high-dimensional data. (ii) Furthermore, the learned explanation masks on the pixel level remain difficult for humans to understand.

theoretic bounds. However, the Merlin-Arthur framework faces significant scalability challenges when applied to high-dimensional real-world data, as explanations based on raw pixels are both computationally difficult to optimize and offer limited human understandability (*cf.* Fig. 1; Wäldchen et al. (2024)).

Concurrently, *Concept Bottleneck Models* (CBMs) have emerged as a powerful framework for interpretable machine learning, structuring predictions through intermediate interpretable concept encodings (Koh et al., 2020; Stammer et al., 2021). Despite their advantages, CBMs typically employ *linear classifiers* on top of concept encoding layers, thereby potentially restricting expressivity and failing on tasks requiring nonlinear interactions among concepts (e.g., XOR problems, counting or permutation invariance, *cf.* Suppl. B) (Mahinpei et al., 2021; Kimura et al., 2024; Lee et al., 2019).

In this work, we combine the best of both worlds by introducing the **Neural Concept Verifier (NCV)**, a novel framework integrating concept-based representations into PVGs in the form of Merlin-Arthur Classifiers. NCV shifts the prover–verifier interaction from the image level to a structured, symbolic concept level, overcoming both the scalability limitations encountered by Merlin-Arthur Classifiers in high-dimensional settings and the expressivity constraints inherent to linear CBMs. Through extensive evaluations on controlled synthetic benchmarks such as CLEVR-Hans, as well as large-scale real-world datasets including CIFAR-100 and ImageNet-1k, we demonstrate that NCV successfully scales PVGs to complex, high-dimensional classification tasks. At the same time, NCV enables verifiable, performant nonlinear classifiers on top of concept extractors, effectively narrowing the interpretability–accuracy gap present in standard, linear CBMs. Lastly, our framework improves robustness to shortcut learning, thereby enhancing the generalizability and trustworthiness of predictions, particularly crucial in high-stakes applications.

Our contributions can be summarised as: (i) We propose *Neural Concept Verifier* (NCV), a framework combining concept-based models with Prover–Verifier Games (PVGs). (ii) NCV scales PVGs to high-dimensional image data by operating on compact concept encodings. (iii) It enables expressive yet interpretable classification via sparse, nonlinear reasoning over concepts. (iv) We validate NCV on synthetic and real-world benchmarks, demonstrating strong accuracy and verifiability. (v) We highlight that NCV improves generalization under spurious correlations, indicating increased robustness.

The remainder of the paper is structured as follows. We begin with a review of related work, highlighting recent developments in the field. We then introduce our proposed Neural Concept Verifier framework, and present its formal description. This is followed by a comprehensive experimental evaluation that investigates key aspects of the framework. Finally, we discuss our findings and conclude the paper.

## 2 BACKGROUND

**Prover-Verifier Games.** PVGs were introduced by Anil et al. (2021) as a game-theoretic framework to encourage learning agents to produce *testable* justifications through interactions between an untrusted prover and a trusted verifier. Their work showed that under suitable game settings, the verifier can learn robust decision rules even when the prover actively attempts to persuade it of arbitrary outputs. Wäldchen et al. (2024) extended this idea to the *Merlin-Arthur Classifier* (MAC), which provides formal interpretability guarantees by bounding the mutual information between the selected features and the ground-truth label. Recently, Kirchner et al. (2024) applied a PVG-inspired approach to improve legibility of Large Language Model (LLM) outputs and Amit et al. (2024) introduced *self-proving models* that leverage interactive proofs to formally verify the correctness of model outputs. PVG-style setups have also been explored in safety-focused learning protocols (Irving et al., 2018; Brown-Cohen et al., 2024; Głuch et al., 2024). These developments reflect a broader trend of utilising multi-agent learning (Pruthi et al., 2022; Schneider & Vlachos, 2024; Du et al., 2024; Nair et al., 2023; Stammer et al., 2024a). Our work builds on this line of research by embedding PVGs into a concept-based classification framework, addressing the dimensionality bottleneck that limits PVGs in high-dimensional settings.

**Concept Representations for Interpretability.** The introduction of Concept Bottleneck Models (CBMs) (Koh et al., 2020; Delfosse et al., 2024), but also concept-based explanations (Kim et al., 2018; Crabbé & van der Schaar, 2022; Poeta et al., 2023; Lee et al., 2025) was an important moment in the growing interest in AI interpretability research. The particular appeal of CBMs lies in the promise of interpretable predictions and a controllable, structured interface for human interactions (Stammer et al., 2021). While initial CBMs relied on fully supervised concept annotations, subsequent research has relaxed this requirement by leveraging pretrained vision-language models like CLIP for concept extraction (Bhalla et al., 2024; Yang et al., 2023; Oikarinen et al., 2023; Panousis et al., 2024; Steinmann et al., 2025), or employing fully unsupervised concept discovery methods (Ghorbani et al., 2019; Stammer et al., 2024b; Schut et al., 2025; Sawada & Nakamura, 2022). Overall, much work has focused on enhancing the concept bottleneck itself—by reducing supervision requirements, mitigating concept leakage, or dynamically expanding the concept space. However, several recent approaches also enrich the classifier component using nonlinear or symbolic predictors, including Concept Embedding Models (Espinosa Zarlenga et al., 2022), concept-based memory reasoning (Debot et al.,

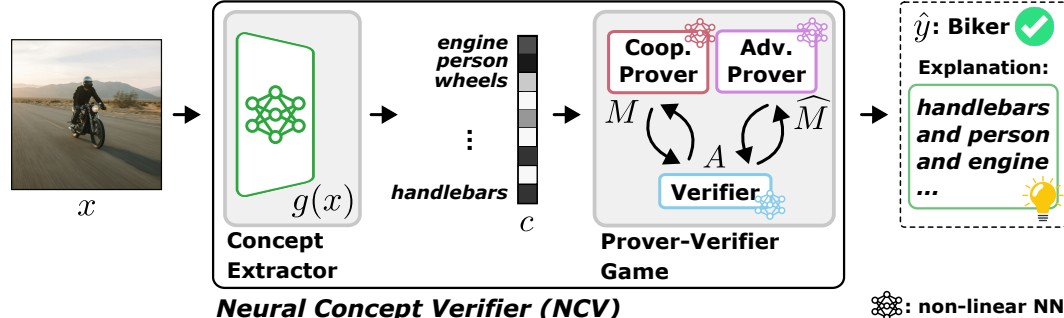

Figure 2: Overview of the Neural Concept Verifier (NCV). The input image is first processed by a concept extractor to produce symbolic concept encodings. A prover–verifier game is then played over these encodings: a cooperative prover selects a sparse concept subset supporting the true class, while an adversarial prover selects misleading concepts. Finally, the nonlinear verifier makes a prediction based only on these selected concepts, ensuring verifiable and robust classification.

2024), neural-symbolic reasoning (Barbiero et al., 2023), and causal concept models (Dominici et al., 2024; De Felice et al., 2025). NCV is complementary to these methods: rather than proposing a new classifier architecture, it wraps any concept-level predictor in a prover–verifier game that enforces sparse, per-sample concept selection and evaluates predictions under competing subsets.

**Shortcut Learning.** Independent of their interpretability, training deep models can often lead to unwanted artifacts and side effects. Shortcut learning describes the problem of models learning to rely on unwanted and unintended features to resolve a task (Geirhos et al., 2020). This is a common problem when training purely deep learning models (Lapuschkin et al., 2019; Schramowski et al., 2020) or neuro-symbolic models (Marconato et al., 2023; Bortolotti et al., 2025), and, if not taken care of, can lead to predictions being right for the wrong reasons (Ross et al., 2017). There have been various methods developed to tackle this problem, from careful dataset curation (Ahmed et al., 2021) to modified model training (Friedrich et al., 2023), *cf.* (Steinmann et al., 2024) for a comprehensive overview. However, these mitigation methods rely on several important assumptions about the training data and potential shortcuts to mitigate, and can affect model performance (Sagawa et al., 2019). While NCV is not specifically designed to mitigate shortcuts, we show that our setup can intrinsically mitigate the impact of shortcuts in the data.

## 3 NEURAL CONCEPT VERIFIER (NCV)

In this section, we introduce the *Neural Concept Verifier* (NCV), a framework that combines concept-based representations with the Merlin–Arthur prover-verifier paradigm (Wäldchen et al., 2024). NCV trains nonlinear classifiers whose predictions provably rely on sparse subsets of high-level concepts, with guarantees formalized through completeness and soundness criteria (*cf.* Suppl. A). This enables scaling PVGs from low-dimensional settings to high-dimensional inputs via concept encodings. In contrast to the original MAC, which operates on raw pixel features and struggles to scale to high-dimensional inputs, NCV performs the prover–verifier interaction directly in concept space. This shift enables optimization on complex datasets and grounds explanations in human-interpretable concepts. NCV consists of two main components (*cf.* Fig. 2): (i) a minimally or weakly supervised *concept extractor* that transforms input data into interpretable concept encodings, and (ii) a nonlinear MAC that selects and verifies sparse concept-based inputs to make final predictions. After providing background notations and a high-level overview of the NCV framework below, we provide detailed descriptions of each main component as well as training and inference details in the following.

### 3.1 PROBLEM SETUP AND NOTATION

Let $\mathcal{X} \in \mathbb{R}^{N \times D}$ denote a dataset of $N$ inputs (e.g., images), each of dimension $D$, and let $\mathcal{Y} \in \{1, \ldots, K\}^N$ be the corresponding class labels for $K$ classes. We assume that the pairs $(x, y)$ are drawn i.i.d. from an unknown data distribution $\mathcal{D}$ over $\mathbb{R}^D \times \{1, \ldots, K\}$, and that $(\mathcal{X}, \mathcal{Y})$ correspond to a finite sample from $\mathcal{D}$. For the verifier, we consider an extended prediction space $\{1, \ldots, K, \bot\}$

with an additional *rejection class* $\perp$, which allows the model to abstain from a decision when uncertain and is crucial for enforcing interpretability guarantees in adversarial setups (Wäldchen et al., 2024). Our overall goal is to learn a model $f : \mathbb{R}^D \to \{1, \ldots, K, \perp\}$ that maps an input $x \in \mathbb{R}^D$ to a prediction $\hat{y} \in \{1, \ldots, K, \perp\}$, where the prediction is based on a small, interpretable subset of high-level concepts.

### 3.2 THE NCV FRAMEWORK

NCV next decomposes $f$ into the following components (*cf.* Fig. 2 for an illustrative overview):

1. A **concept extractor** $g : \mathcal{X} \to \mathcal{C}$, which maps each input $x$ to a high-level concept encoding $\mathbf{c} \in \mathbb{R}^C$, where $C \in \mathbb{N}$ is the number of discovered concepts.

2. A pair of **prover agents** $M, \widehat{M} : \mathcal{C} \to \{0, 1\}^C$ that produce sparse binary masks selecting $m$ concepts each. $M$ (Merlin, *cooperative Prover*) aims to help classification; $\widehat{M}$ (Morgana, *adversarial Prover*) aims to mislead, which is crucial for overall robustness.

3. A nonlinear **verifier** (Arthur) $A : \mathbb{R}^C \to \mathcal{Y}$, which predicts a label based only on the masked concepts.

The three agents are trained jointly, where the interaction between these agents encourages the verifier to rely only on robust, informative concept features. Let us now provide details on these individual components.

### 3.3 CONCEPT EXTRACTION

The concept extractor $g : \mathcal{X} \to \mathbb{R}^C$ transforms raw input data into interpretable, high-level concept representations, where each dimension corresponds to a semantically meaningful concept. The resulting concept encoding $\mathbf{c} \in \mathbb{R}^C$ serves as the input to the PVG. The value of $C$ is determined entirely by the concept extractor (e.g., vocabulary size or number of discovered concepts) and is treated as a fixed input dimensionality for the subsequent prover-verifier interaction.

While conceptually simple, a careful combination of PVGs and concept-based models is necessary. The concept extractor must satisfy three key requirements: interpretability, expressiveness, and modularity. Concepts should correspond to human-understandable features that can serve as meaningful explanations, the concept space should capture sufficient information for the target task without creating information bottlenecks, and the extractor should operate independently of the prover-verifier components. Unlike traditional concept extractor approaches that enforce sparsity constraints directly on $\mathbf{c}$, our framework delegates sparsity to the prover-verifier interaction. This allows the concept space to remain dense and expressive while achieving interpretable sparsity through downstream concept selection. Further, NCV can accommodate different supervision paradigms for concept extraction: supervised methods that leverage predefined concept vocabularies, self-supervised methods that exploit multi-modal correspondences (e.g., vision-language alignment), or unsupervised methods that discover latent conceptual structures can all be used as concept extractors.

Overall, NCV requires that $g$ produces consistent, interpretable encodings while maintaining sufficient information for accurate classification. In our evaluations, we instantiate NCV's concept extractor via the recent unsupervised and object-centric NCB framework (Stammer et al., 2024b) and the multi-modal, CLIP-based SpLiCE (Bhalla et al., 2024) approach. Operating in concept space rather than raw input space provides: (i) scalability through dimensionality reduction and (ii) explanations based on human-interpretable concepts.

### 3.4 VERIFIABLE CLASSIFICATION VIA THE MERLIN-ARTHUR CLASSIFIERS

The second core component of NCV is a verifiable classifier of the Merlin-Arthur setup (Wäldchen et al., 2024) originally inspired by Interactive Proof Systems (Goldwasser et al., 1985). This setup generally formalizes the idea of proving that a classification decision is supported by a sparse and informative set of features.

In NCV, specifically, two competing provers, **Merlin** and **Morgana**, select concept subsets either to support or mislead classification, respectively. The verifier, Arthur, then makes predictions based

solely on these masked concepts without knowledge of the prover's intent. Formally, each prover outputs a sparse binary mask with $m$ active entries, producing selected subsets $S = M(\mathbf{c}) \odot c$ and $\widehat{S} = \widehat{M}(\mathbf{c}) \odot \mathbf{c}$, where $\mathbf{c}$ is the concept encoding from extractor $g$, and $\odot$ denotes element-wise masking. Notably, all three agents represent differentiable models, with Arthur specifically representing a nonlinear model.

This interactive setup enables two key metrics: completeness and soundness. Writing $S = M(\mathbf{c}) \odot \mathbf{c}$ for Merlin's cooperative subset and $\widehat{S} = \widehat{M}(\mathbf{c}) \odot \mathbf{c}$ for Morgana's adversarial subset, we define

$$\text{Completeness} \;=\; \mathbb{P}_{(x,y)\sim\mathcal{D}}\big[A(S) = y\big], \tag{1}$$

$$\text{Soundness} \;=\; \mathbb{P}_{(x,y)\sim\mathcal{D}}\big[A(\widehat{S}) \in \{y, \bot\}\big], \tag{2}$$

where $\bot$ denotes the rejection class. Intuitively, completeness measures how often Arthur can recover the true label from Merlin's sparse, helpful concepts, while soundness measures how often Arthur can avoid committing to a wrong label under Morgana's misleading subset, either by staying correct or abstaining (*cf.* Suppl. A for the theoretical interpretation). Overall, these components in NCV encourage the verifier to base its decisions on inspectable, sparse concept subsets, leading to information-theoretic guarantees (*cf.* Sec. A.2).

Sparsity has recently become central in concept-based models, as large concept spaces require sparse predictions for interpretability. Prior work (Bhalla et al., 2024; De Santis et al., 2025) achieves this by regularizing the concept space itself, restricting the number of active concepts before training the classifier. In contrast, NCV keeps the concept space fully expressive and enforces sparsity only in the concepts passed to the classifier. At inference, Arthur predicts solely from the masked concepts selected by Merlin, ensuring sparse, interpretable predictions without limiting the richness of the concept space.

### 3.5 Training and Inference

The training step of NCV incorporates updating only the parameters of the three agents $M$, $\widehat{M}$, and $A$ as $g$ represents a pretrained model that is subsequently frozen upon NCV's multi-agent training step. Thus, the three agents are jointly trained by optimizing a three-agent game, which encourages Arthur to rely on concepts selected by Merlin, while being robust to potentially misleading concepts selected by Morgana. Given a concept encoding $\mathbf{c} \in \mathbb{R}^C$, label $y \in \mathcal{Y}$ and cross-entropy function $CE(\cdot, \cdot)$, we define:

- **Merlin's loss:** $L_M = CE(A(\mathcal{S}), y)$, where $\mathcal{S}$ is the sparse concept subset selected by Merlin. This loss encourages Arthur to classify correctly based on Merlin's input.

- **Morgana's loss:** $L_{\widehat{M}} = CE(A(\widehat{\mathcal{S}}), y)$, where $\widehat{\mathcal{S}}$ is Morgana's adversarial concept subset. Here, the loss is interpreted as the classifier's inability to be misled by deceptive inputs[1].

Overall, Arthur's loss combines both objectives with a hyperparameter $\gamma \in \mathbb{R}_{\geq 0}$, controlling the emphasis on predictive performance (*completeness*) versus robustness (*soundness*):

$$L_A = (1 - \gamma)\, L_M + \gamma\, L_{\widehat{M}}, \tag{3}$$

In detail, the three agents are updated jointly in a two-phase min-max optimization. First, the prover agents are updated where Merlin minimizes $L_M$ and Morgana maximizes $L_{\widehat{M}}$; then Arthur is updated by minimizing $L_A$ on the sparse selected concepts chosen by the provers. This scheme incentivizes Arthur to base its predictions on informative, task-relevant, and verifiably robust concept subsets.

At inference time, only the cooperative prover $M$ is used to select a sparse subset of concepts, based on the input's concept encoding. The verifier $A$ then predicts a label or rejects based solely on this selected subset.

Overall, by integrating concept-extractor modules and leveraging the Merlin–Arthur framework, NCV emphasizes faithfulness[2] and interpretability while preserving nonlinear modeling capabilities,

---

[1]In practice, the CE loss of Morgana is a slightly modified CE loss (*cf.* Sec. A.4).

[2]Our notion of 'faithfulness' is defined in Sec. A.3.

and shifts the min–max optimization into a lower-dimensional concept space, improving efficiency, scalability, and stability in high-dimensional settings, a common challenge in min–max optimization for deep learning (Mescheder et al., 2018; Nagarajan & Kolter, 2017). A more detailed discussion of completeness, soundness, and their information-theoretic interpretation in NCV is provided in Suppl. A.

# 4 EXPERIMENTAL EVALUATIONS

In this section, we present a comprehensive evaluation of Neural Concept Verifier (NCV) on both synthetic and real-world high-dimensional image datasets. We evaluate based on two instantiations of NCV that utilize different concept extractors: a CLIP-based extractor and the Neural Concept Binder (NCB). We assess predictive performance and interpretability across multiple datasets, compare against several baselines and examine scalability and robustness against shortcut learning.

Our evaluation is structured around the following research questions: **(Q1)** Does shifting Prover-Verifier Games (PVGs) to concept-encodings via NCV lead to performative classifiers on high-dimensional synthetic and real-world images (*i.e.*, high completeness and soundness)? **(Q2)** Does NCV reduce the "interpretability-accuracy gap" in the context of CBMs? **(Q3)** Does NCV allow for more detailed explanations over pixel-based PVGs? Finally, **(Q4)** Can training via NCV reduce shortcut learning?

## 4.1 EXPERIMENTAL SETUP

**Datasets.** We investigate NCV on CLEVR-Hans3 and CLEVR-Hans7 (Stammer et al., 2021), synthetic benchmarks derived from CLEVR (Johnson et al., 2017) that capture complex object compositions and include visual shortcuts. CLEVR-Hans3 features three compositional classes, while CLEVR-Hans7 increases the complexity to seven, with all images rendered at $128 \times 128$ pixels. The training and validation sets contain spurious correlations between attributes and labels (e.g., gray cubes linked to a specific class), which are absent in the test set, making the datasets well-suited for studying shortcut behavior. Models that exploit such correlations often fail under the decorrelated test distribution. We first report results on non-confounded versions of these datasets, where feature distributions are consistent across splits, and later return to the confounded versions for shortcut mitigation. To assess scalability and generalization to natural images, we additionally evaluate on ImageNet-1k (Deng et al., 2009) with 1.2M high-resolution images across 1,000 classes (resized to $224 \times 224$ pixels), and on CIFAR-100 (Krizhevsky, 2009) with 60,000 low-resolution $32 \times 32$ images across 100 fine-grained categories. Lastly, we perform experiments on COCOLogic (Steinmann et al., 2025), a recent benchmark combining real-world images with complex, compositional class rules.

**Baseline Models.** We compare our framework against several representative baselines, with training details provided in Suppl. C. As a strong but non-interpretable baseline, we use a standard ResNet-18 (He et al., 2016) for evaluations on CLEVR-Hans, and a ResNet-50 for CIFAR-100, CO-COLogic and ImageNet-1k, each trained end-to-end on raw images. We further evaluate a pixel-based MAC (Wäldchen et al., 2024) (denoted as *Pixel-MAC*), an instantiation of the Prover-Verifier Game in which the verifier is initialized from a pretrained ResNet-18, while both provers (Merlin and Morgana) are U-Net models (Ronneberger et al., 2015) that output continuous feature-importance masks over the input image. These masks are discretized using Top-$k$ selection to define the features visible to the verifier, and all agents are jointly fine-tuned; the resulting explanations (*i.e.*, certificates) correspond to masks in pixel space (see (Wäldchen et al., 2024) for further details). Lastly, we compare to a vanilla Concept Bottleneck Model (Koh et al., 2020) (denoted as *CBM*), where a linear classifier predicts from concept features extracted by either NCB (Stammer et al., 2024b) for CLEVR-Hans or SpLiCE (Bhalla et al., 2024) for CIFAR-100, ImageNet-1k and COCOLogic. In addition, we include a nonlinear CBM variant (CBM+MLP), which replaces the linear classifier by a two-layer MLP operating on the same concept encodings; this baseline isolates the effect of a more expressive concept-level predictor without changing the underlying concept extractor.

**NCV Instantiations.** For CLEVR-Hans3 and CLEVR-Hans7, we instantiate NCV with NCB (Stammer et al., 2024b) as the concept extractor, using models pretrained on CLEVR (Johnson et al., 2017). A permutation-invariant Set Transformer (Lee et al., 2019) serves as the verifier (Arthur) to process the unordered NCB encodings. The provers (Merlin and Morgana) are independent Set Transformers

Table 1: NCV delivers high predictive performance and soundness through verifiable, concept-based reasoning evaluated via completeness and soundness. We report completeness and soundness scores for ResNet, Pixel-MAC, CBM, and NCV across synthetic (CLEVR-Hans3, CLEVR-Hans7) and real-world (CIFAR-100, ImageNet-1k, COCOLogic) datasets. NCV matches or outperforms baselines in completeness in most settings, while offering strong soundness guarantees.

| Model | Feature Space | Completeness (Accuracy) | Soundness (Robustness) | Completeness (Accuracy) | Soundness (Robustness) | Completeness (Accuracy) | Soundness (Robustness) |
|---|---|---|---|---|---|---|---|
| | | **CIFAR-100** | | **ImageNet-1k** | | **COCOLogic** | |
| ResNet-50 | pixel space | $81.45_{\pm 0.60}$ | n/a | $76.01_{\pm 0.02}$ | n/a | $65.80_{\pm 3.41}$ | n/a |
| CBM (nonlin.) | SpLiCE | $79.29_{\pm 0.42}$ | n/a | $69.02_{\pm 0.38}$ | n/a | $70.09_{\pm 0.56}$ | n/a |
| Pixel-MAC | pixel space | $15.27_{\pm 4.78}$ | $96.31_{\pm 4.12}$ | $35.06_{\pm 3.20}$ | $99.65_{\pm 0.26}$ | $42.57_{\pm 3.13}$ | $97.70_{\pm 0.61}$ |
| CBM | SpLiCE | $75.42_{\pm 0.04}$ | n/a | $\mathbf{68.59_{\pm 0.01}}$ | n/a | $58.84_{\pm 0.09}$ | n/a |
| **NCV** (ours) | CLIP-Sim | $\mathbf{83.32_{\pm 0.28}}$ | $\mathbf{99.99_{\pm 0.01}}$ | $67.04_{\pm 0.16}$ | $\mathbf{99.94_{\pm 0.02}}$ | $\mathbf{75.42_{\pm 3.21}}$ | $\mathbf{97.87_{\pm 0.47}}$ |
| | | **CLEVR-Hans3** | | **CLEVR-Hans7** | | | |
| ResNet-18 | pixel space | $97.87_{\pm 0.24}$ | n/a | $98.71_{\pm 0.24}$ | n/a | | |
| Pixel-MAC | pixel space | $96.59_{\pm 0.72}$ | $99.99_{\pm 0.01}$ | $97.61_{\pm 0.38}$ | $99.88_{\pm 0.28}$ | | |
| CBM | NCB | $95.44_{\pm 0.08}$ | n/a | $89.12_{\pm 0.12}$ | n/a | | |
| **NCV** (ours) | NCB | $\mathbf{98.92_{\pm 0.32}}$ | $\mathbf{100.00_{\pm 0.00}}$ | $\mathbf{97.89_{\pm 0.31}}$ | $\mathbf{100.00_{\pm 0.00}}$ | | |

that take the full concept-slot encodings as input and output a sparse mask of 12 active concepts for the verifier. All components are jointly trained with the Adam optimizer (Kingma & Ba, 2014). Further details and ablations are provided in Suppl. D. For ImageNet-1k, CIFAR-100 and COCOLogic, we use a CLIP-based concept extractor (Radford et al., 2021), following the approach of SpLiCE (Bhalla et al., 2024) to compute image–text similarity scores with a fixed concept vocabulary. Unlike SpLiCE, which performs per-sample optimization, our method (denoted as *CLIP-Sim*) retains the full activation vector and delegates concept selection to the provers, avoiding expensive inference-time optimization and enabling scalability. Here, the verifier and both provers are two-layer MLPs; the provers output sparse masks of 32 concepts per example. All modules are trained with Adam. Additional details and ablations are provided in Suppl. D and Suppl. E, including the effect of varying the mask size and the weighting parameter $\gamma$.

**Metrics.** All methods are evaluated for completeness and, where applicable, soundness (Sec. 3.4). Here, completeness coincides with standard classification accuracy when Arthur is evaluated on Merlin's cooperative subsets $S$, while soundness is the probability that Arthur either predicts the correct label or abstains when evaluated on Morgana's adversarial subsets $\widehat{S}$. We use 20 random seeds for CLEVR-Hans and 10 for ImageNet-1k, CIFAR-100 and COCOLogic, reporting mean and standard deviation across all seeds. For CLEVR-Hans shortcut learning, we additionally report a separate *shortcut robustness* metric in Table 2: the validation–test gap, i.e., the difference between validation accuracy on a confounded split and test accuracy on a non-confounded split; smaller gaps indicate better generalization and reduced shortcut reliance. This shortcut robustness metric is independent of soundness and does not involve the prover–verifier game.

## 4.2 EVALUATIONS

**Scaling PVGs to High Dimensions (Q1).** In our first evaluation, we examine whether shifting the Prover–Verifier Game (PVG) to concept encodings enables NCV to scale to high-dimensional image domains while achieving strong performance in terms of completeness and soundness. We hereby compare NCV against two key baselines: (1) a black-box ResNet classifier (ResNet-18 for CLEVR-Hans and ResNet-50 for CIFAR-100, ImageNet-1k and COCOLogic), and (2) Pixel-MAC, a nonlinear PVG model operating in raw pixel space. Tab. 1 summarizes results across synthetic (CLEVR-Hans3, CLEVR-Hans7) and real-world (CIFAR-100, ImageNet-1k, COCOLogic) benchmarks. Each model's feature space is indicated for clarity. On the synthetic CLEVR-Hans benchmarks, we observe that NCV consistently achieves the highest completeness scores, surpassing Pixel-MAC and even ResNet-18 on CLEVR-Hans3, while also attaining perfect soundness. This demonstrates that NCV not only matches or exceeds the performance of strong black-box classifiers but also certifiable decision-making. Pixel-MAC performs well in these settings but falls slightly short in completeness and cannot match NCV's zero-error soundness.

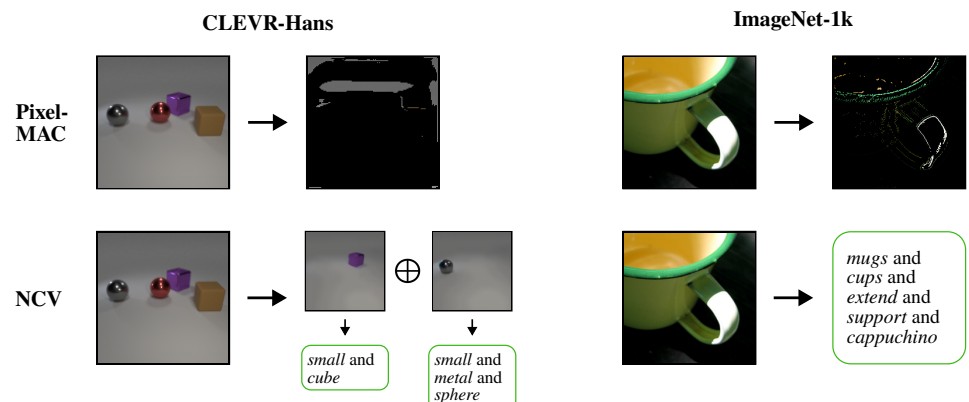

Figure 3: Comparison of explanations from NCV *vs.* Pixel-MAC. **(top)** Merlin–Arthur training on pixel space yields uninformative masks. **(bottom)** MAC on concept encodings via NCV translates into combinations of high-level concepts and, in turn, in an interpretable prediction. The conjunction "and" in these outputs simply concatenates the individual concepts selected by Merlin and does not denote a learned logical AND operator.

On the more challenging real-world datasets, Pixel-MAC either fails entirely or performs poorly. In contrast, NCV successfully scales to these datasets, achieving superior completeness and near-perfect soundness. Notably, NCV surpasses ResNet-50 even in raw accuracy for CIFAR-100 and COCOLogic, providing both higher predictive performance while retaining interpretability. In summary, NCV generalizes well across domains: it scales beyond the limitations of pixel-based PVGs, delivers competitive accuracy even on large-scale and complex datasets, and retains soundness throughout. Additional comparisons with the DCR baseline (Barbiero et al., 2023) using the same 10k-concept CLIP-Sim vocabulary are reported in the supplements (*cf.* Tab. 4 in Suppl. C), where NCV substantially outperforms DCR on CIFAR-100 and COCOLogic-10 and DCR failing to scale to ImageNet-1k in our setup. Overall, these findings affirm that shifting PVGs to concept space enables interpretable classifiers whose decisions can be evaluated via completeness and soundness, while remaining performant and scalable in high-dimensional synthetic and real-world visual environments. We therefore answer Q1 affirmatively.

**Narrowing the Interpretability–Accuracy Gap (Q2).** In Tab. 1, we further examine whether NCV can overcome a central limitation of standard Concept Bottleneck Models (CBMs): the interpretability–accuracy gap resulting from their use of constrained linear classifiers (*cf.* Suppl. B for a discussion). Related to this, we observe that NCV consistently narrows and in some cases even closes this gap, while maintaining high completeness and soundness across all evaluated datasets. Specifically, on CLEVR-Hans3, the baseline CBM trails the opaque ResNet-18 by over 2 percentage points in completeness, whereas NCV matches or exceeds the ResNet's performance while retaining perfect soundness. The benefit is even more pronounced on CLEVR-Hans7: CBM underperforms ResNet-18 by nearly 10 percentage points, while NCV narrows the gap to just 1 percentage point. This trend persists on real-world datasets. On CIFAR-100, NCV outperforms the base CBM and even slightly exceeds ResNet-50's performance. This is even more pronounced on COCOLogic, where NCV outperforms both the base CBM and ResNet-50 by a large margin. As the additional benefits of a nonlinear classifier are quite small on ImageNet-1k, the additional training complexity of NCV results in a slightly worse performance compared to the base CBM there. Overall, NCV improves over linear CBMs in both accuracy and robustness, especially on tasks requiring complex concept reasoning. At the same time, it can match or even surpasses the accuracy of the opaque ResNet models, demonstrating that interpretable, concept-level reasoning via Prover–Verifier Games can deliver competitive performance, without sacrificing the completeness and soundness criteria. Complementing these results, the nonlinear CBM+MLP baseline reduces the accuracy gap between CBMs and ResNets, but still falls short of NCV on most datasets and does not provide per-sample, prover–verifier style explanations or guarantees. We therefore answer Q2 affirmatively.

**More Detailed Explanations (Q3)** We next investigate the resulting explanations produced by our NCV framework, with a focus on explanatory clarity. Since our goal is to improve over classic vision-

Table 2: Shortcut robustness on CLEVR-Hans3 and CLEVR-Hans7. We report validation accuracy on a shortcut-confounded split, test accuracy on a clean split, and the resulting validation–test gap ("shortcut robustness", lower is better) across models trained with varying amounts of clean data.

| Ratio Non-Conf. (Samples) | Model | CLEVR-Hans3 | | | CLEVR-Hans7 | | |
|---|---|---|---|---|---|---|---|
| | | Val Acc (w/ shortcut) | Test Acc (w/o shortcut) | Val-Test Gap ($\downarrow$) | Val Acc (w/ shortcut) | Test Acc (w/o shortcut) | Val-Test Gap ($\downarrow$) |
| 0% | CBM (lin.) | $95.65 \pm 0.09$ | $90.54 \pm 0.09$ | 5.11 | $90.37 \pm 0.10$ | $85.27 \pm 0.15$ | **5.10** |
| | CBM (non-lin.) | $98.70 \pm 0.32$ | $\mathbf{95.04} \pm 0.96$ | **3.66** | $98.09 \pm 0.24$ | $90.69 \pm 1.17$ | 7.40 |
| | NCV | $\mathbf{99.44} \pm 0.15$ | $94.21 \pm 1.41$ | 5.23 | $\mathbf{98.38} \pm 0.18$ | $\mathbf{92.23} \pm 0.67$ | 6.15 |
| 1% (105) | CBM (lin.) | $96.28 \pm 0.16$ | $91.03 \pm 0.31$ | 5.25 | $90.74 \pm 0.12$ | $85.41 \pm 0.17$ | 5.33 |
| | CBM (non-lin.) | $99.10 \pm 0.27$ | $94.84 \pm 0.98$ | 4.26 | $98.17 \pm 0.17$ | $92.65 \pm 1.31$ | 5.52 |
| | NCV | $\mathbf{99.37} \pm 0.18$ | $\mathbf{97.11} \pm 0.98$ | **2.26** | $\mathbf{98.19} \pm 0.24$ | $\mathbf{94.68} \pm 0.64$ | **3.51** |
| 5% (525) | CBM (lin.) | $95.38 \pm 0.37$ | $93.34 \pm 0.51$ | 2.04 | $90.37 \pm 0.15$ | $86.37 \pm 0.18$ | 4.00 |
| | CBM (non-lin.) | $98.41 \pm 0.55$ | $96.13 \pm 0.71$ | 2.28 | $98.32 \pm 0.22$ | $95.19 \pm 0.80$ | 3.13 |
| | NCV | $\mathbf{99.59} \pm 0.19$ | $\mathbf{98.88} \pm 0.37$ | **0.71** | $\mathbf{98.47} \pm 0.24$ | $\mathbf{96.24} \pm 0.71$ | **2.23** |
| 20% (2100) | CBM (lin.) | $95.67 \pm 0.28$ | $93.46 \pm 0.23$ | 2.21 | $89.93 \pm 0.29$ | $87.21 \pm 0.31$ | 2.72 |
| | CBM (non-lin.) | $99.15 \pm 0.21$ | $98.09 \pm 0.51$ | 1.06 | $98.21 \pm 0.29$ | $97.00 \pm 0.49$ | 1.21 |
| | NCV | $\mathbf{99.37} \pm 0.28$ | $\mathbf{98.82} \pm 0.67$ | **0.55** | $\mathbf{98.63} \pm 0.13$ | $\mathbf{97.74} \pm 0.28$ | **0.89** |

based Prover–Verifier Games, we compare against pixel-level MAC explanations. Fig. 3 illustrates a qualitative example from both the CLEVR-Hans3 (*cf.* Fig. 4 for more examples) and ImageNet-1k datasets. Notably, under Pixel-MAC, the Prover–Verifier setup operates directly on pixels, yielding broad, diffuse explanation masks that often cover entire objects or irrelevant background regions, arguably providing limited insight regarding which exact features drive the verifier's final decision. In contrast, NCV leverages its internal concept encodings to isolate sparse, high-level concepts that are consistently associated with a class decision under the prover–verifier interaction, recovering the class rule for CLEVR-Hans (i.e., small cube and small metal sphere) and providing a meaningful concept explanation for the coffee-mug class of ImageNet-1k.[3] For ImageNet-1k, the mask size is set to 32 concepts, but for clarity we visualize only the top 5 most frequent concepts across 32 samples. In the ImageNet-1k setting, our CLIP/SpLiCE-based concept vocabulary is derived from LAION-based vocabulary (Schuhmann et al., 2021) (see Suppl. E for more details), which can occasionally yield overly generic or noisy concept labels (e.g., "extend"), and thus limits the quality of the resulting textual explanations.

Overall, these examples highlight that NCV offers higher-level, semantically meaningful explanations rather than fine-grained pixel masks, and that concept-level PVGs yield interpretable decisions whose supporting concept subsets can be evaluated via completeness and soundness even for complex, high-dimensional data. This leads us to answer Q3 affirmatively.

**Mitigating Shortcut Learning (Q4)** Lastly, to assess whether NCV can mitigate shortcut learning in image classification, we train models on different versions of CLEVR-Hans3 and CLEVR-Hans7 with varying ratios of clean samples (*i.e.*, without shortcut) in the training and validation sets. We then measure validation accuracy with shortcuts and test accuracy on a held-out, clean data split. This setup allows us to track both predictive performance and robustness to shortcut learning. Tab. 2 reports results for three model types: a linear CBM, a nonlinear CBM, and our instantiation of NCV using NCB as concept extractor. We observe that while NCV achieves the highest test accuracy among all models in the 0% clean data setting, it still exhibits a sizeable validation-test gap, indicating a strong influence of the underlying shortcuts. As the amount of clean samples is progressively increased, test accuracy and test-validation gap improves across all models. However, NCV consistently achieves the highest test accuracy in every setting, and its validation–test gap decreases more rapidly than for either CBM variant. This trend indicates that NCV is not only better at leveraging clean supervision when available, but is also more robust to shortcut learning. Together, these results demonstrate

---

[2]For CLEVR-Hans, NCV uses NCB's object-centric slots to reconstruct objects from Merlin's concept selections; for ImageNet-1k, it visualizes CLIP-based high-level semantic concepts.

[3]The availability of object-level concepts in NCV depends on the underlying concept extractor. For CLEVR-Hans, we use NCB, which provides such object-based explanations.

that concept-level Prover–Verifier Games in NCV encourage models to rely on robust, task-relevant features, making NCV more resilient to shortcut learning, even with limited amounts of clean data.

## 5    DISCUSSION

Overall, our results show that shifting Prover–Verifier Games (PVGs) to the concept level yields a powerful and scalable framework for verifiable, interpretable classification. By operating on symbolic concept embeddings, NCV avoids the computational cost of per-sample inference in pixel space, yet matches or surpasses pixel-based baselines in both completeness and soundness. It reduces the performance gap typical of Concept Bottleneck Models (CBMs), achieving parity with opaque models on synthetic tasks and even surpassing them on natural images. Additionally, concept-level outputs offer concise, human-readable explanations. Finally, NCV exhibits a resilience to spurious correlations, generalizing from confounded training splits and closing the generalization gap with minimal available clean data.

That said, NCV has several limitations. Its effectiveness depends on the quality of the underlying concept extractor: noisy or entangled concept spaces can reduce both accuracy and human understandability. The increased training complexity introduced by the three-agent PVG setup also results in greater computational cost and training instability, e.g., compared to linear CBMs. Moreover, when using pretrained models like CLIP for concept discovery, NCV inherits their biases and inconsistencies to some extent (Birhane et al., 2021; Gehman et al., 2020; Bhalla et al., 2024). Finally, recent work (Debole et al., 2025) shows that such concept spaces can diverge from expert semantics, even when yielding strong downstream performance.

## 6    CONCLUSION

In this work, we have introduced the Neural Concept Verifier (NCV), a unified framework that brings together Prover–Verifier Games and concept-level representations for interpretable classification at scale. Through extensive experiments on CLEVR-Hans, CIFAR-100, ImageNet-1k, and COCOLogic, we have shown that NCV achieves high completeness and soundness, reduces the interpretability–accuracy gap of concept bottleneck models, delivers detailed concept-based explanations, and effectively mitigates shortcut learning. Thus, NCV paves the way for deploying trustworthy and transparent models in domains where both predictive performance and verifiability are essential.

Future work should explore how concept encodings can be integrated into alternative PVG-style setups, where structured representations may improve performance or reduce communication overhead. It is also promising to investigate applications beyond vision, such as natural language processing and structured data, where interpretable verification may be equally valuable. At the optimization level, our current setup does not train Merlin and Morgana end-to-end on discrete, binarized masks; developing more stable optimization schemes for discrete concept selection could further strengthen the framework and the provers themselves. Finally, while existing information-theoretic guarantees, such as those introduced by Wäldchen et al. (2024), focus on binary classification under specific assumptions, extending such guarantees to high-dimensional, multi-class settings remains an important open direction for formal interpretability at scale.

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

# Supplementary Materials

## A    Theoretical Guarantees and Relation to Merlin–Arthur Classifiers

Neural Concept Verifier (NCV) builds on the Merlin–Arthur classifier (MAC) framework of Wäldchen et al. (2024), which provides information-theoretic interpretability guarantees in a binary classification setting. In this appendix, we briefly recall these guarantees and explain how they apply in our concept-based setting. We do *not* prove new theorems here; rather, we instantiate existing results and make explicit which assumptions are required and where our claims remain empirical.

### A.1    Merlin–Arthur guarantees in the binary case

We briefly recall the guarantees for Merlin–Arthur classifiers in the original binary setting of Wäldchen et al. (2024), focusing on the quantities that appear in our NCV discussion: average precision, mutual information, completeness, soundness, asymmetric feature correlation, and relative success rate.

**Setup.**    We consider a two-class data space $D = (\mathcal{X}, \mathcal{D}, \ell)$ with label $L = \ell(X) \in \{-1, 1\}$, where $X \sim \mathcal{D}$ and the class-conditional distributions are $\mathcal{D}_l := \mathcal{D}(\cdot \mid L = l)$. Features are partial objects $z \in \mathcal{D}_p$ (*e.g.*, subsets of pixels) that can be contained in a data point $x$; we write $z \subseteq x$. A feature selector $M$ (Merlin or Morgana) maps $x$ to a feature $M(x) \subseteq x$, and Arthur is a classifier $A$ that predicts in $\{-1, \perp, 1\}$, where $\perp$ denotes abstention.

**Average precision and mutual information.**    Given a feature $z$ and a second data point $x \sim \mathcal{D}$, the precision of $z$ is the probability that $x$ has the same label as a reference point $x'$ that exhibits $z$:

$$\Pr(z; x') := \mathbb{P}_{x \sim \mathcal{D}}\big[\ell(x) = \ell(x') \mid z \subseteq x\big].$$

For a feature selector $M$, the *average precision* is the expected precision of the features it selects:

$$\Pr_{\mathcal{D}}(M) := \mathbb{E}_{x' \sim \mathcal{D}}\Big[\mathbb{P}_{x \sim \mathcal{D}}\big[\ell(x) = \ell(x') \mid M(x') \subseteq x\big]\Big]. \tag{4}$$

This quantity bounds the average conditional entropy of the class given Merlin's features and thus the mutual information. Writing $H_b$ for the binary entropy and $H(\cdot)$, $I(\cdot; \cdot)$ for entropy and mutual information, Wäldchen et al. (2024) show

$$\mathbb{E}_{x' \sim \mathcal{D}}\big[I_{x \sim \mathcal{D}}\big(\ell(x); M(x') \subseteq x\big)\big] \geq H_{x \sim \mathcal{D}}\big(\ell(x)\big) - H_b\big(\Pr_{\mathcal{D}}(M)\big). \tag{5}$$

Thus, as $\Pr_{\mathcal{D}}(M) \to 1$, the binary entropy term $H_b(\Pr_{\mathcal{D}}(M)) \to 0$ and Merlin's features carry almost all available label information in this binary setting.

**Idealised min–max guarantee (optimal players).**    In the idealised setting, Arthur and Morgana are assumed to play optimally against a fixed Merlin. Define the error set

$$E_{M, \widehat{M}, A} := \Big\{x \in \mathcal{X} \,\big|\, A(M(x)) \neq \ell(x) \ \lor \ A(\widehat{M}(x)) = -\ell(x)\Big\},$$

*i.e.*, points where Merlin fails to convince Arthur of the correct label, or Morgana successfully forces an incorrect (non-abstaining) prediction. The min–max error of Merlin is

$$\varepsilon_M := \min_A \max_{\widehat{M}} \Pr_{x \sim \mathcal{D}}[x \in E_{M, \widehat{M}, A}].$$

Theorem 2.7 in Wäldchen et al. (2024) states that if $\varepsilon_M$ is small, there exists a subset $\mathcal{X}' \subseteq \mathcal{X}$ of mass at least $1 - \varepsilon_M$ such that, restricted to the induced data space $D' = (\mathcal{X}', \mathcal{D}', \ell)$, Merlin achieves perfect precision:

$$\Pr_{\mathcal{D}'}(M) = 1 \quad \Rightarrow \quad H_{x', x \sim \mathcal{D}'}\big(\ell(x') \mid M(x') \subseteq x\big) = 0.$$

In words: if an optimally trained Arthur–Morgana pair can almost never disagree with Merlin, then on almost all of the data space Merlin's features determine the label uniquely.

**Realistic players: completeness, soundness, AFC, and relative strength.** For high-dimensional data, exhaustive search for an optimal Morgana is not feasible. The analysis is therefore relaxed to *realistic* (*e.g.*, neural) players and expressed in terms of:

- **Completeness**
$$\min_{l \in \{-1,1\}} \mathbb{P}_{x \sim \mathcal{D}_l}\big[A(M(x)) = \ell(x)\big] \ \geq \ 1 - \varepsilon_c,$$
*i.e.*, Merlin's features let Arthur classify correctly with high probability in each class.

- **Soundness**
$$\max_{l \in \{-1,1\}} \mathbb{P}_{x \sim \mathcal{D}_l}\big[A(\widehat{M}(x)) = -\ell(x)\big] \ \leq \ \varepsilon_s,$$
*i.e.*, Morgana almost never forces a confidently wrong prediction; on her subsets, Arthur either stays correct or abstains (predicts $\perp$).

- **Asymmetric Feature Correlation (AFC)** $\kappa$, which measures how strongly a set of features can be *concentrated* in a few points of one class but *spread out* across many points of the other class. Large AFC allows Merlin to use globally uninformative features (appearing equally often in both classes) in a way that still yields high completeness and soundness.

- **Class imbalance** $B$, which upper-bounds how skewed the class prior can be (formally, a bound on the ratio of class masses).

- **Relative success rate** $\alpha$ of Morgana, which compares how often Morgana can find a convincing feature in the *wrong* class to how often Merlin can do so in the *correct* class, restricted to points that contain at least one feature Merlin uses successfully:

$$\alpha := \min_{l \in \{-1,1\}} \frac{\mathbb{P}_{x \sim \mathcal{D}_{-l}}\big[A(\widehat{M}(x)) = l \mid x \in F_l^*\big]}{\mathbb{P}_{x \sim \mathcal{D}_l}\big[A(M(x)) = l \mid x \in F_l^*\big]},$$

where $F_l^*$ is the set of points that contain a feature Merlin uses to convince Arthur of class $l$. Intuitively, $\alpha$ is large if Morgana's search procedure is at least as powerful as Merlin's.

Under these conditions, Wäldchen et al. (2024) prove that completeness, soundness, AFC, class imbalance, and relative success jointly lower-bound the average precision:

$$\Pr_{\mathcal{D}}(M) \ \geq \ 1 - \varepsilon_c - \frac{\kappa \alpha^{-1} \varepsilon_s}{1 - \varepsilon_c + \kappa \alpha^{-1} B^{-1} \varepsilon_s}. \tag{6}$$

Combining the bound on $\Pr_{\mathcal{D}}(M)$ in equation 6 with the mutual-information inequality equation 5, we obtain:

$$\mathbb{E}_{x'}\big[I\big(\ell(x); M(x') \subseteq x\big)\big] \ \geq \ H\big(\ell(x)\big) - H_b\left(1 - \varepsilon_c - \frac{\kappa \alpha^{-1} \varepsilon_s}{1 - \varepsilon_c + \kappa \alpha^{-1} B^{-1} \varepsilon_s}\right).$$

For *balanced* datasets ($B \approx 1$), bounded AFC ($\kappa = O(1)$), and a reasonably strong Morgana ($\alpha = O(1)$), high completeness ($\varepsilon_c \ll 1$) and soundness ($\varepsilon_s \ll 1$) therefore imply that Merlin's features carry almost all label information in this binary setting.

## A.2 INSTANTIATION FOR CONCEPT-BASED MODELS

NCV applies the same prover–verifier game as Merlin–Arthur classifiers, but in a *concept space* rather than pixel space. For an input $x \in \mathcal{X}$, a concept extractor $g$ produces an encoding $\mathbf{c} = g(x) \in \mathbb{R}^C$. We interpret each concept index $j \in \{1, \ldots, C\}$ as a feature. Merlin and Morgana are implemented as provers $M, \widehat{M} : \mathbb{R}^C \to \{0,1\}^C$ that output sparse binary masks $M(\mathbf{c}), \widehat{M}(\mathbf{c})$ with at most $m$ active entries. Arthur then predicts using the masked encodings

$$S = M(\mathbf{c}) \odot \mathbf{c} \quad \text{and} \quad \widehat{S} = \widehat{M}(\mathbf{c}) \odot \mathbf{c}.$$

To connect this to the binary setting of Section A.1, consider a fixed class $k \in \{1, \ldots, K\}$ and the associated one-vs-rest binary task with label $Y_k \in \{-1, 1\}$ (class $k$ vs. all others). For this binary subproblem we view:

- the concept indices $j \in \{1, \ldots, C\}$ as features,
- Merlin's selector $M$ as a map that sends $\mathbf{c} = g(x)$ to a subset $M(\mathbf{c}) \subseteq \{1, \ldots, C\}$,
- Arthur as a classifier that predicts in $\{-1, \bot, 1\}$ from the masked encoding $M(\mathbf{c}) \odot \mathbf{c}$.

Let $\mathrm{Pr}_{\mathcal{D}}^{(k)}(M)$ denote the *average precision* of Merlin's concept features on this one-vs-rest task for class $k$, defined analogously to equation 4:

$$\mathrm{Pr}_{\mathcal{D}}^{(k)}(M) := \mathbb{E}_{x' \sim \mathcal{D}}\Big[\mathbb{P}_{x \sim \mathcal{D}}\big[Y_k(x) = Y_k(x') \mid M(x') \subseteq x\big]\Big].$$

Applying the mutual-information inequality equation 5 to the binary label $Y_k$ and selector $M$ yields

$$\mathbb{E}_{x' \sim \mathcal{D}}\big[I_{x \sim \mathcal{D}}\big(Y_k(x); \; M(x') \subseteq x\big)\big] \; \geq \; H(Y_k) - H_b\big(\mathrm{Pr}_{\mathcal{D}}^{(k)}(M)\big). \tag{7}$$

Thus, if $\mathrm{Pr}_{\mathcal{D}}^{(k)}(M)$ is close to 1, Merlin's sparse concept subsets for class $k$ carry almost all information about $Y_k$.

The analysis of Section A.1 further relates $\mathrm{Pr}_{\mathcal{D}}^{(k)}(M)$ to *observable* completeness and soundness on this binary subproblem, under three additional assumptions:

- a bounded *Asymmetric Feature Correlation* (AFC) parameter $\kappa_{\mathrm{concept}}$ in concept space. Formally, each one-vs-rest subproblem for class $k$ has its own AFC parameter $\kappa_k$; for notational simplicity we write $\kappa_{\mathrm{concept}}$ for a uniform upper bound $\kappa_{\mathrm{concept}} \geq \kappa_k$ over all classes.
- a bounded *class imbalance* $B_k$ for the one-vs-rest task;
- a non-degenerate *relative success rate* $\alpha_k$ of Morgana, meaning that Morgana's search procedure over concept subsets is roughly as powerful as Merlin's. In the ideal full-search setting one has $\alpha_k = 1$; in NCV we use symmetric neural architectures for Merlin and Morgana as heuristic evidence that $\alpha_k$ is close to 1, but we do not estimate it explicitly.

Under these conditions, the precision bound equation 6 applies class-wise: for each $k$ one obtains

$$\mathrm{Pr}_{\mathcal{D}}^{(k)}(M) \; \geq \; 1 - \varepsilon_c^{(k)} - \frac{\kappa_{\mathrm{concept}}(\alpha_k)^{-1}\varepsilon_s^{(k)}}{1 - \varepsilon_c^{(k)} + \kappa_{\mathrm{concept}}(\alpha_k)^{-1}(B_k)^{-1}\varepsilon_s^{(k)}},$$

where $\varepsilon_c^{(k)}$ and $\varepsilon_s^{(k)}$ are the completeness and soundness errors of NCV on the one-vs-rest problem for class $k$. Combining this with equation 7 yields a lower bound on the mutual information between the class-$k$ label and Merlin's concept subsets.

In NCV we view these results as an *idealised* description of the concept-level prover–verifier game:

- On binary tasks and under the AFC and relative-strength assumptions above, high completeness and soundness imply that Merlin's sparse concept subsets are highly informative about the label in the sense of equation 7.
- In our high-dimensional, multi-class experiments we do not estimate the classwise parameters $(\varepsilon_c^{(k)}, \varepsilon_s^{(k)}, \alpha_k, B_k, \kappa_{\mathrm{concept}})$ explicitly. We therefore interpret the Merlin–Arthur theory as a *theoretical lens* for NCV, not as a quantitative certification for each dataset; empirically reported completeness and soundness should be read in this light (*cf.* the discussion in Sec. 6).

### A.3 NOTION OF FAITHFULNESS

Throughout the paper, when we say that NCV produces *faithful* explanations, we mean this in the information-theoretic sense inherited from Merlin–Arthur and adapted to concept space:

1. For an input $x$ with true class $k$, Merlin produces a sparse concept subset $S(x) = M(\mathbf{c}) \odot \mathbf{c}$ (with $\mathbf{c} = g(x)$) such that, with high probability over $x \sim \mathcal{D}_k$, Arthur can correctly predict $k$ from $S(x)$ alone, *i.e.*, completeness for the corresponding one-vs-rest task is high.

2. An adversarial prover cannot find alternative concept subsets that force Arthur into a wrong prediction; at worst, Arthur abstains via the rejection class $\perp$. This is soundness in the sense of Section A.1.

3. Under bounded concept-space AFC, bounded class imbalance, and a reasonably strong Morgana (relative success rate $\alpha_k$ not too small), the precision bound of Section A.1 links the class-wise average precision $\mathrm{Pr}_{\mathcal{D}}^{(k)}(M)$ to the completeness and soundness errors $\varepsilon_c^{(k)}$ and $\varepsilon_s^{(k)}$. In particular, when these errors are small and the parameters $(\kappa_{\mathrm{concept}}, \alpha_k, B_k)$ are well-behaved, the bound guarantees that $\mathrm{Pr}_{\mathcal{D}}^{(k)}(M)$ is close to 1, and the mutual-information inequality equation 7 then implies that Merlin's sparse concept subsets for class $k$ carry near-maximal information about $Y_k$. In our experiments we report completeness and soundness empirically, but we do not attempt to estimate the resulting lower bounds on $\mathrm{Pr}_{\mathcal{D}}^{(k)}(M)$.

This is explicitly *not* a causal guarantee: NCV does not prove that the concepts are causally sufficient for the task. It only ensures that, relative to the given concept representation and under the assumptions above, the sparse subsets Merlin selects are as informative and robust as possible under the prover–verifier game.

## A.4 USE OF THE REJECTION CLASS AND TRAINING OBJECTIVE

Finally, we clarify how the rejection class is used during training, since it is crucial for enforcing soundness in the sense of Sec. A.1. Arthur outputs logits $A(S) \in \mathbb{R}^{K+1}$ over $K$ classes plus a rejection class $\perp$. Denote by $p_y$ and $p_\perp$ the corresponding softmax probabilities for the true class $y$ and for the rejection class, respectively.

Given a concept encoding $\mathbf{c} = g(x)$ and Merlin/Morgana subsets $S = M(\mathbf{c}) \odot \mathbf{c}$ and $\widehat{S} = \widehat{M}(\mathbf{c}) \odot \mathbf{c}$, we use the following losses:

- **Merlin loss**
$$L_M = -\log p_y,$$
i.e., standard cross-entropy w.r.t. the true class based on Merlin's subset.

- **Morgana loss (soundness).** In our implementation, Morgana's loss is realised by operating directly on the logits for the true class and the rejection class. Let $z = A(\widehat{S}) \in \mathbb{R}^{K+1}$ denote Arthur's logits on Morgana's subset, and write $z_y$ and $z_\perp$ for the components corresponding to the true class $y$ and the rejection class $\perp$, respectively.
First, we define a modified target label $\tilde{y}$ that switches to the rejection class whenever Arthur already prefers $\perp$ over $y$ on $\widehat{S}$:
$$\tilde{y} = \begin{cases} y, & \text{if } z_y \geq z_\perp, \\ \perp, & \text{if } z_\perp > z_y. \end{cases}$$

We then define Morgana's loss as

$$L_{\widehat{M}} = \underbrace{\mathrm{CE}(z, \tilde{y})}_{\text{cross-entropy on } \tilde{y}} + \underbrace{\frac{1}{B} \sum_{i=1}^{B} \left( -\log\left(1 + \exp(-|z_y^{(i)} - z_\perp^{(i)}|)\right)\right)}_{\text{stabilising log-sum-exp term}},$$

where $B$ is the batch size and $z_y^{(i)}, z_\perp^{(i)}$ are the logits for example $i$ in the batch. The first term encourages Arthur, on Morgana's subsets, to either predict the true class $y$ or abstain (predict $\perp$) whenever $\perp$ is already preferred. The second term acts as a smooth regulariser that keeps the difference $|z_y - z_\perp|$ in a numerically stable range: it discourages pushing these logits arbitrarily far apart and thus stabilises gradients for both Arthur and Morgana, while still allowing Arthur to separate $y$ and $\perp$ when beneficial.

Arthur's overall loss is a convex combination

$$L_A = (1 - \gamma)L_M + \gamma L_{\widehat{M}},$$

with $\gamma \in [0, 0.5]$ trading off completeness and soundness. In practice, all losses are averaged over the batch (mean reduction).

Although Merlin's and Morgana's concept selections are ultimately discrete, both provers are trained via continuous masks. Concretely, each prover outputs real-valued scores $m_{\text{cont}}, \widehat{m}_{\text{cont}} \in \mathbb{R}^C$, which are used as soft masks to form

$$S_{\text{soft}} = m_{\text{cont}} \odot \mathbf{c}, \qquad \widehat{S}_{\text{soft}} = \widehat{m}_{\text{cont}} \odot \mathbf{c}.$$

When updating the provers (Merlin by gradient descent on $L_M$, Morgana by gradient ascent on $L_{\widehat{M}}$), Arthur is frozen and we feed $S_{\text{soft}}$ and $\widehat{S}_{\text{soft}}$ into $A$, so that gradients from Arthur's losses flow back into the continuous mask parameters.

After these prover updates, we discretise the masks using a top-$m$ operator: for each input, we set the $m$ entries of largest magnitude in $m_{\text{cont}}$ (respectively $\widehat{m}_{\text{cont}}$) to 1 and all others to 0, obtaining hard masks $M(\mathbf{c}), \widehat{M}(\mathbf{c}) \in \{0, 1\}^C$. Arthur is then updated (gradient descent on $L_A$) using the corresponding hard-masked encodings $S$ and $\widehat{S}$.

This alternating scheme — continuous masks for gradient flow in the provers and hard top-$m$ masks for Arthur's update — implements a stable and practical min–max training procedure for NCV. The explicit use of the rejection class $\perp$ ensures that soundness measures robustness to *adversarial* concept selections (Arthur is not allowed to be confidently wrong) and aligns with the Merlin–Arthur framework instantiated in concept space.

### A.5 LIMITATIONS OF THE THEORETICAL GUARANTEES

For completeness, we also spell out the main limitations of the Merlin–Arthur guarantees when applied to NCV's high-dimensional, multi-class, concept-based setting.

First, the original theory is formulated for binary classification. Our instantiation in Section A.2 uses a one-vs-rest reduction to obtain class-wise guarantees on the mutual information $I(Y_k; M(x') \subseteq x)$ for each $k$, but it does not provide a direct statement about the joint $K$-class decision or about the final $\arg\max$ prediction over all classes.

**High-dimensional sparsity and feature reuse.** A central limitation arises from the extreme sparsity of Merlin's and Morgana's selections in our high-dimensional concept spaces. In the original Merlin–Arthur setting, features are typically small, localized structures (*e.g.*, image patches) and the experiments are conducted on relatively low-dimensional, small-scale datasets. In this regime, many inputs share the same features, so the event $\{M(x') \subseteq x\}$ has non-negligible probability and the average precision $\Pr_{\mathcal{D}}(M)$ in Eq. 4 can be meaningfully interpreted and estimated from finite samples. In NCV, Merlin and Morgana select small subsets of a large concept vocabulary (*e.g.*, $m \ll C$ with $C$ in the thousands), and in practice the exact subsets $M(x')$ and $\widehat{M}(x')$ are often highly specific to each input. As a result, the precise event $\{M(x') \subseteq x\}$ may have very low probability under $\mathcal{D}$, and empirical estimates of $\Pr_{\mathcal{D}}^{(k)}(M)$ become unstable in finite samples. For this reason, we do not attempt to estimate average precision or the resulting mutual-information lower bounds numerically in our experiments. Instead, we use completeness and soundness as observable proxies and treat the precision/MI guarantees as an idealised, distribution-level description of the behaviour that NCV is designed to promote, rather than as directly calibrated finite-sample certificates.

**Class imbalance in one-vs-rest reductions.** In large-$K$ settings such as ImageNet-1k, the one-vs-rest subproblems for each class $k$ are highly imbalanced (the positive prior is approximately $1/K$). In the precision bound of Sec. A.2, this is reflected in the imbalance parameter $B_k$, whose large value makes the resulting lower bound on $\Pr_{\mathcal{D}}^{(k)}(M)$ more conservative: small soundness errors $\varepsilon_s^{(k)}$ are penalised more strongly relative to completeness $\varepsilon_c^{(k)}$. This provides an additional reason why we do not attempt to compute numerical mutual-information lower bounds in our experiments, and instead interpret the Merlin–Arthur theory qualitatively as a guiding framework.

**Relative success and adversary class.** Finally, the robustness interpretation of soundness in our setting is tied to the particular adversarial prover class we train in practice (a neural network $\widehat{M}$

with a specific architecture and loss). In the Merlin–Arthur framework, this dependence is captured abstractly via the relative success rate $\alpha$, which measures how powerful Morgana is compared to Merlin on those points where Merlin can provide convincing evidence. In NCV we use symmetric neural architectures for $M$ and $\widehat{M}$ as heuristic evidence that $\alpha_k$ is not tiny, but we do not attempt to verify or optimise $\alpha_k$ beyond this design choice. As a result, our empirical soundness estimates should be interpreted as robustness against this trained adversary class, rather than against an arbitrary worst-case adversary over all admissible subsets. This is fully in line with the practical use of the Merlin–Arthur framework in Wäldchen et al. (2024), where the general theory is instantiated and evaluated with specific neural implementations of Merlin, Morgana, and Arthur.

Overall, we see the Merlin–Arthur framework as providing a rigorous *idealised model* for the kind of behaviour NCV is designed to encourage: high completeness, robustness to adversarial concept subsets, and information-rich sparse explanations in concept space. In our experiments, we use completeness and soundness as observable proxies for these properties, but we do not claim fully certified guarantees beyond the stated assumptions.

## B    WHY LINEAR CLASSIFIERS FALL SHORT IN CBMS

While linear classifiers are generally considered to be interpretable, these models are not suited to solve arbitrarily complex problems. A linear classifier is only able to capture linear relationships between inputs and output features and cannot model complex, non-linear relationships. In the context of CBMs, this problem is usually tackled by utilizing a linear classifier to predict the output based on the detected concepts. The concepts themselves can be detected using non-linear models, and only the classification based on these concepts is done with a linear model. However, this is not always sufficient, as there are also simple examples where non-linear relationships between concepts and the output exist, for example thresholds detection (three out of five symptoms need to be present to indicate an illness) or multiplicative effects (crop yield is the result of a multiplicative relationship between rain and fertility).

To illustrate the problem in a simple experimental setup, let us assume we have a dataset of simple shapes and every image contains between one and four of these shapes. The shapes are either a square or a circle and either orange or blue. We consider two simple classification scenarios for this dataset.

- **XOR:** This setting classification follows the traditional XOR problem: We want to classify images that contain either an orange square or a blue circle as class one and all other images as class two.
- **Counting:** This setting includes object counting and illustrates that even for classification based on a single attribute, a linear layer can be insufficient. Here, we want to classify all images with exactly one blue shape as class one, and all other images as class two.

We evaluate a linear layer and a simple MLP on this toy dataset. To further simplify things, we assume that our concept encoder is able to perfectly detect the concepts in the image, thus providing for each element the information whether there is an object and if so, its shape and color. We randomly generate 5000 samples of the dataset and train the models on a train split of 80% and evaluate on the remaining 20%. The MLP has one hidden layer of size 16 and uses ReLU activation functions.

The results of this evaluation are shown in Tab. 3. In both scenarios, the linear classification layer is not able to solve the task, despite the deceptively simple relationship between concepts and output classes. On the other hand, the MLP achieves close to perfect accuracy on both settings.

So far, we have argued that not every task can be solved with a CBM and a linear classification layer. However, this is not

Table 3: A linear prediction layer cannot solve *XOR* or *counting*. Even with the assumption of a perfect concept encoder, the linear layer fails.

| Model | XOR (Acc) | Counting (Acc) |
|---|---|---|
| Linear Layer | $0.766 \pm 0.011$ | $0.677 \pm 0.006$ |
| MLP | $0.953 \pm 0.053$ | $0.982 \pm 0.015$ |

entirely accurate. In principle, any task can be solved linearly—provided that we define the right *linear-sufficient* concepts. For instance, in the XOR setting, detecting the concepts "orange square and no blue circle" and "blue circle and no orange square" would allow a linear classifier to solve the

task. Similarly, in the counting task, introducing a concept such as "exactly one blue object" would make linear classification trivial.

That said, the assumption that such sufficient concepts are always available is not realistic. First, designing or discovering these concepts often makes concept detection considerably more difficult. Second, as concepts become increasingly specific and compositional, they tend to lose interpretability. Finally, requiring tailored concepts for every individual task does not scale. Returning to our example, the concept "exactly one blue object" might help with task two but is essentially useless for task one.

Taken together, this illustrates why relying solely on linear classifiers in CBMs is often impractical. To address such cases, non-linear classifiers should also be considered.

## C EXPERIMENTAL DETAILS: BASELINES

In this section, we provide training details for the considered baselines: *ResNet-18*, *ResNet-50*, *Pixel-MAC*, *CBM* (linear and nonlinear variants), and *DCR*.

### C.1 RESNET-18 AND RESNET-50

We initialize the framework with a pretrained ResNet-18 model and employ the Adam optimizer across all experiments. On the CLEVR datasets, the model is trained with a batch size of 128 for 30 epochs using a learning rate of $10^{-4}$ and weight decay of $10^{-4}$, repeated across 20 random seeds with early stopping based on validation loss.

On CIFAR-100, we use a ResNet-50 trained for 100 epochs with a learning rate of $10^{-4}$, weight decay of $10^{-5}$, and a batch size of 128, averaged over 10 random seeds with early stopping. The ResNet-50 baseline on ImageNet is evaluated directly using pretrained PyTorch (Paszke et al., 2019) weights without further finetuning. For COCOLogic, a ResNet-50 is trained for 300 epochs with a batch size of 256, learning rate of $10^{-4}$, and weight decay of $10^{-2}$, again averaged over 10 random seeds with early stopping.

In the Pixel-MAC setup, a separate ResNet-18 is trained under the same configuration as above but with a reduced learning rate of $10^{-5}$, while keeping the batch size, weight decay, and early stopping criterion unchanged. All Pixel-MAC results are obtained from these ResNet-18 checkpoints.

### C.2 PIXEL-MAC

In this setup, we apply Merlin-Arthur training on pixel space by utilizing the pretrained ResNet-18 models as classifiers and U-Net architectures for both Merlin and Morgana. Throughout all experiments, we employ the Adam optimizer for both classifier and feature selector optimization, with $\gamma = 0.5$ to ensure high soundness.

For the CLEVR datasets, we train with a batch size of 128 for 40 epochs, using a learning rate of $10^{-5}$ and weight decay of $10^{-6}$ for the classifier optimization. The U-Net architectures are trained with a learning rate of $10^{-4}$, weight decay of $10^{-5}$ and an L1 penalty coefficient of 0.1. We set the mask size to 1500, meaning that the U-Nets select a subset of 1500 pixels per sample (out of $128 \times 128$ pixels).

For CIFAR-100, we reduce the batch size to 64 and train for 100 epochs. The classifier is optimized with a learning rate of $10^{-5}$ and weight decay of $10^{-4}$. Both Merlin and Morgana are trained using a learning rate of $10^{-3}$ and a reduced mask size of 32 pixels. We use an L1 penalty of 0.01 and apply early stopping across 10 random seeds.

For ImageNet and COCOLogic, we further reduce the batch size to 32 due to memory constraints and train for 80 epochs using the same learning rates and hyperparameters as in the CIFAR-100 setting. The mask size is set to 1000 pixels per image. As with CIFAR-100, early stopping is applied across 10 random seeds, and training is initialized from the pretrained ResNet-18 backbone.

## C.3 CBM with Linear Classifier

Next, we present the implementation details for the CBM baseline, where a linear classifier operates on concept features obtained from the concept extractor.

For the CLEVR datasets, we train a linear classifier on concepts extracted by the Neural Concept Binder. The training process employs a batch size of 128, a learning rate of $10^{-3}$, and weight decay of $10^{-4}$. The model is trained for 60 epochs on CLEVR-Hans3 and 30 epochs on CLEVR-Hans7, using early stopping based on validation loss, repeated across 20 different random seeds.

For CIFAR-100 and ImageNet, we train a linear classifier on sparse SpLiCE encodings using a dictionary size of 10,000. In both cases, we use a batch size of 4096 and train for 250 epochs with early stopping, a learning rate of $10^{-3}$, and no weight decay. A hidden layer with 512 units is used, and an L1 penalty of 0.2 is applied within SpLiCE to encourage sparsity in the concept representations. All results are averaged over 10 random seeds.

## C.4 DCR Baseline Implementation

For the additional nonlinear CBM comparison in Sec. 4, we implemented Deep Concept Reasoning (DCR; Barbiero et al. 2023) following the official `torch-explain` library. Our implementation uses the same experimental setup as NCV to ensure a fair comparison.

**Architecture.** We employ a ResNet-18 backbone (pretrained weights disabled for consistency) followed by two DCR-specific modules: (1) a `ConceptEmbedding` layer that maps backbone features to concept predictions $c_{pred} \in [0, 1]^C$ and learned concept embeddings $c_{emb} \in \mathbb{R}^{C \times d}$, and (2) a `ConceptReasoningLayer` that learns class-specific logic rules in Disjunctive Normal Form (DNF) over the concept embeddings to produce class predictions.

**Concept Supervision.** We supervise the concept predictions using the same CLIP-Sim similarity vectors as NCV. Since these are cosine similarities in $[-1, 1]$, we apply a linear normalization $c_{truth} = (\text{sim} + 1)/2$ to map them to $[0, 1]$ for BCE loss compatibility.

**Training.** Following the official DCR tutorial, we optimize a joint loss $\mathcal{L} = \mathcal{L}_{concept} + \lambda \cdot \mathcal{L}_{task}$, where both terms use binary cross-entropy (BCE). The task loss uses one-hot encoded labels, as the `ConceptReasoningLayer` outputs per-class probabilities (not logits). We train with Adam optimizer for 100 epochs.

**Hyperparameter Sweep.** For each dataset, we performed a grid search over:

- Vocabulary size: $n \in \{1000, 3000, 10000\}$
- Concept embedding dimension: $d \in \{8, 16, 32\}$
- Learning rate: $\eta \in \{10^{-3}, 10^{-4}\}$
- Task loss weight: $\lambda \in \{0.5, 1.0\}$

This resulted in $3 \times 3 \times 2 \times 2 = 36$ configurations per dataset, totaling over 100 training runs. For COCOLogic-10, we report balanced accuracy due to class imbalance.

**Results.** Tab. 4 summarizes the best accuracies achieved by DCR compared to NCV at 10k vocabulary size. DCR successfully trains on CIFAR-100 and COCOLogic-10 but fails to complete training on ImageNet-1k within reasonable time ($> 3$ days per epoch). In contrast, NCV's sparsity-inducing mechanism enables efficient scaling to all three datasets under the same 10k-concept space.

# D NCB-based Neural Concept Verifier Experiments

In the following, we provide details on NCB-based NCV, experimental evaluations as well as additional evaluations.

Table 4: Comparison between DCR and NCV at 10k-concept vocabulary size on CLIP-Sim. NCV metrics correspond to completeness (accuracy). For COCOLogic-10, we report balanced accuracy due to class imbalance.

| Dataset | DCR (acc. %) | **NCV** (ours) (completeness, %) |
|---|---|---|
| CIFAR-100 | 51.76 | **83.32** |
| ImageNet-1k | n/a (timeout) | **67.04** |
| COCOLogic-10 (bal.) | 38.61 | **75.42** |

## D.1 PRETRAINING

Before training NCV, we first pretrain the models without the feature selectors. The corresponding results for the pretraining are shown in Tab. 5, where we evaluate on 20 random seeds. These pretrained models are then used as initialization for the subsequent NCV training. For the pretraining, we use a Set Transformer with two stacked multi-head attention blocks, a hidden dimension of 128 and four attention heads. We use a batch size of 128, 30 epochs and the Adam optimizer with a learning rate of $10^{-3}$ for both datasets, applying early stopping based on validation loss.

Table 5: Pretraining results on the CLEVR-Hans3 and CLEVR-Hans7 datasets without shortcuts

| **CLEVR-Hans3** | | **CLEVR-Hans7** | |
|---|---|---|---|
| Val. Accuracy | Test Accuracy | Val. Accuracy | Test Accuracy |
| $99.02 \pm 0.31$ | $98.13 \pm 0.37$ | $98.08 \pm 0.24$ | $97.83 \pm 0.25$ |

## D.2 NCV TRAINING

For the experiments presented in our main results in Tab. 1, the experimental details for both datasets are as follows:

**Model Architecture.** The verifier is implemented as a pretrained Set Transformer consisting of two stacked multi-head attention blocks with hidden dimension 128, four attention heads, and layer normalization. Merlin and Morgana are implemented as independent neural networks, each parameterized by a Set Transformer with two stacked attention blocks with hidden dimensions 256, four attention heads, and layer normalization. The provers receive the full concept slot matrix as input and output a sparse selection mask with exactly 12 nonzero entries (out of 64 total features), indicating the active blocks provided to the verifier.

**Training Details.** All components are jointly trained using the Adam optimizer with a learning rate of $10^{-3}$ and weight decay of $10^{-4}$. Models are trained for 50 epochs and a batch size of 512 is used throughout. For the Merlin and Morgana provers, a hard selection constraint is enforced, limiting the number of selected concepts to a fixed budget of 12 block-encodings per sample. To ensure high soundness, we set $\gamma = 0.5$, giving equal weight to both feature selector losses in the total loss computation. We train our models using 20 random seeds.

**Extended Results.** Additionally, we evaluated the NCV framework with varying mask sizes and an alternative model architecture for the feature selectors. The results are presented in Tab. 6 for the CLEVR-Hans3 dataset and Tab. 7 for the CLEVR-Hans7 dataset, where we evaluate both the validation set and the test set. The alternative architecture implements a MLP with two hidden layers and ReLU activation functions for the feature selectors, while maintaining a pretrained Set Transformer as the classifier across all experiments. Our results reveal that the Set Transformer feature selector consistently outperforms the MLP feature selector on the test set, particularly with smaller mask sizes such as 4 and 6. Furthermore, this configuration maintains high completeness (>96%) and soundness (>99%), even with a reduced number of selected features.

Table 6: Completeness and soundness on the CLEVR-Hans3 dataset without shortcuts for different mask sizes and feature selector architectures. The highlighted values are used for Table 1.

| Mask Size | Feature Selector | Validation | | Test | |
|---|---|---|---|---|---|
| | | Completeness | Soundness | Completeness | Soundness |
| 4 | Set Transformer | $98.35 \pm 0.31$ | $99.88 \pm 0.28$ | $97.69 \pm 0.63$ | $99.82 \pm 0.23$ |
| | MLP | $96.51 \pm 1.18$ | $99.81 \pm 0.27$ | $95.54 \pm 1.37$ | $99.85 \pm 0.28$ |
| 6 | Set Transformer | $98.71 \pm 0.52$ | $99.87 \pm 0.13$ | $98.11 \pm 0.62$ | $99.88 \pm 0.19$ |
| | MLP | $96.21 \pm 0.89$ | $99.96 \pm 0.03$ | $94.78 \pm 1.24$ | $99.97 \pm 0.06$ |
| 12 | Set Transformer | $99.20 \pm 0.11$ | $100.00 \pm 0.00$ | $\mathbf{98.92 \pm 0.32}$ | $\mathbf{100.00 \pm 0.00}$ |
| | MLP | $99.28 \pm 0.11$ | $99.98 \pm 0.06$ | $98.89 \pm 0.21$ | $99.99 \pm 0.07$ |

Table 7: Completeness and soundness on the CLEVR-Hans7 dataset without shortcuts for different mask sizes and feature selector architectures. The highlighted values are used for Table 1.

| Mask Size | Feature Selector | Validation | | Test | |
|---|---|---|---|---|---|
| | | Completeness | Soundness | Completeness | Soundness |
| 4 | Set Transformer | $96.69 \pm 1.28$ | $99.93 \pm 0.09$ | $96.71 \pm 1.37$ | $99.91 \pm 0.09$ |
| | MLP | $92.63 \pm 1.24$ | $99.89 \pm 0.12$ | $92.71 \pm 1.31$ | $99.87 \pm 0.13$ |
| 6 | Set Transformer | $97.32 \pm 0.42$ | $99.98 \pm 0.02$ | $97.14 \pm 0.51$ | $99.98 \pm 0.02$ |
| | MLP | $95.43 \pm 1.48$ | $99.88 \pm 0.13$ | $95.12 \pm 1.48$ | $99.86 \pm 0.14$ |
| 12 | Set Transformer | $98.13 \pm 0.11$ | $100.00 \pm 0.00$ | $\mathbf{97.89 \pm 0.31}$ | $\mathbf{100.00 \pm 0.00}$ |
| | MLP | $97.41 \pm 1.07$ | $99.99 \pm 0.03$ | $97.01 \pm 0.93$ | $99.99 \pm 0.04$ |

### D.3 EXPLANATIONS

Here, we present supplementary examples of explanations generated by both Pixel-MAC and NCV on the CLEVR-Hans3 dataset in Fig. 4. These results further substantiate our claim that NCV provides significantly more transparent and interpretable explanations compared to the pixel-based PVG baseline.

## E EXPERIMENTAL DETAILS FOR CLIP-BASED NCV

In the following section, we present the implementation details of CLIP-based NCV training.

### E.1 PRETRAINING

Once more, before starting with the actual NCV training, we first pretrain the models without the provers (Merlin and Morgana). The corresponding results for the pretraining are shown in Tab. 8. As textual concept descriptions $T$, we used the top 10,000 most frequent one- and two-word phrases from LAION (Schuhmann et al., 2021) captions, following the setup of Bhalla et al. (2024). For pretraining the verifier, we use a two-layer multilayer perceptron (MLP) with a hidden dimension of 512 and GELU activations (Hendrycks & Gimpel, 2016) on CIFAR-100, ImageNet-1k, and COCOLogic. On CIFAR-100 and ImageNet, training uses a batch size of 4096 and a learning rate of $10^{-4}$, with dropout (0.3), weight decay of $10^{-4}$, and early stopping (patience 10). On COCOLogic, we instead train for 100 epochs with a batch size of 512, learning rate of $10^{-4}$, and weight decay of $10^{-2}$, using a learning-rate scheduler (plateau, patience 5, factor $10^{-3}$, minimum learning rate $10^{-6}$) and no early stopping. All pretraining is conducted without provers, and the resulting verifiers are used to initialize the CLIP-based NCV training.

### E.2 NCV TRAINING

For the experiments presented in our main results in Tab. 1, we detail the training setup separately for CIFAR-100 and ImageNet.

| Class Rules | Pixel-MAC | Neural Concept Verifier |
|---|---|---|

Figure 4: Comparison of explanations from NCV *vs.* Pixel-MAC for CLEVR-Hans3 images of all three classes. **(a)** Merlin–Arthur training on pixel space yields uninformative masks. **(b)** NCV provides clear explanations by highlighting object features corresponding to the class rule. The single-object images are reconstructions from the respective slots selected by Merlin (prover).

Table 8: Pretraining accuracy of the verifier (without provers) for CLIP-based NCV on CIFAR-100, COCOLogic and ImageNet-1k.

| Dataset | Accuracy (%) |
|---|---|
| CIFAR-100 | 85.96 |
| COCOLogic | 81.39 |
| ImageNet-1k | 77.07 |

**Model Architecture.** The verifier (Arthur) is initialized as the pretrained two-layer multilayer perceptron (MLP) described above. Merlin and Morgana are implemented as independent neural networks, each parameterized by a two-layer MLP with hidden dimension 512 and ReLU activations. Both provers receive the full concept activation vector as input and output a sparse selection mask indicating the active concepts that are passed to the verifier.

**CIFAR-100.** For CIFAR-100, all components are trained jointly for 100 epochs using the Adam optimizer. We set the verifier learning rate to $10^{-4}$, and use $5 \times 10^{-4}$ for both Merlin and Morgana. A batch size of 256 is used throughout. Weight decay is set to $0.1$, and a hard mask size of 32 concepts is enforced per input. To incentivize sparse masks, an L1 penalty of $0.1$ is applied to the provers. A learning rate scheduler (plateau-based) is employed with a patience of 5, minimum learning rate of $10^{-6}$, and decay factor of $0.001$. Early stopping is disabled, and all results are averaged over 10 random seeds.

**ImageNet.** The ImageNet setup mirrors CIFAR-100 in most aspects. We again train for 100 epochs with a batch size of 256, using the same learning rates for verifier ($10^{-4}$) and provers ($5 \times 10^{-4}$), mask size of 32 features, and L1 penalty ($0.1$). Weight decay is reduced to $0.005$ to improve generalization. The same learning rate scheduler and seed setup are used as in the CIFAR-100 experiments.

**COCOLogic.** Training on COCOLogic follows the CIFAR-100 configuration with minor adjustments: models are trained for 100 epochs with a batch size of 512, verifier learning rate of $10^{-4}$, and prover learning rates of $5 \times 10^{-4}$. We use a weight decay of $0.01$, a mask size of 32, and an L1 penalty of

Table 9: Ablation results for our method on CIFAR-100 with varying mask sizes. We report mean $\pm$ std over 10 seeds.

| Mask Size | Completeness Train | Completeness Validation | Soundness Train | Soundness Validation |
|---|---|---|---|---|
| 4 | $74.97_{\pm 4.45}$ | $69.32_{\pm 2.66}$ | $99.85_{\pm 0.07}$ | $99.85_{\pm 0.09}$ |
| 8 | $87.42_{\pm 2.60}$ | $71.20_{\pm 22.43}$ | $99.95_{\pm 0.03}$ | $99.95_{\pm 0.03}$ |
| 16 | $94.08_{\pm 1.46}$ | $81.82_{\pm 0.50}$ | $99.96_{\pm 0.03}$ | $99.97_{\pm 0.02}$ |
| 64 | $97.65_{\pm 0.47}$ | $84.01_{\pm 0.31}$ | $100.00_{\pm 0.00}$ | $100.00_{\pm 0.00}$ |

Table 10: Ablation results for our method on ImageNet with varying mask sizes. We report mean $\pm$ std over 10 seeds.

| Mask Size | Completeness Train | Completeness Validation | Soundness Train | Soundness Validation |
|---|---|---|---|---|
| 4 | $59.30_{\pm 0.34}$ | $55.96_{\pm 0.30}$ | $99.81_{\pm 0.03}$ | $99.83_{\pm 0.06}$ |
| 8 | $64.84_{\pm 0.41}$ | $60.98_{\pm 0.30}$ | $99.94_{\pm 0.01}$ | $99.94_{\pm 0.03}$ |
| 16 | $68.85_{\pm 0.34}$ | $64.60_{\pm 0.12}$ | $99.96_{\pm 0.03}$ | $99.96_{\pm 0.03}$ |
| 64 | $73.35_{\pm 0.39}$ | $69.03_{\pm 0.18}$ | $99.97_{\pm 0.01}$ | $99.97_{\pm 0.02}$ |

0.1 on the provers. As with CIFAR-100, early stopping is disabled, and learning rate scheduling and seed averaging remain unchanged.

### E.3   PROVER AND VERIFIER ARCHITECTURES

In all experiments, the choice of prover and verifier architectures is guided by the structure of the concept representation. For unordered, slot-based concept encodings (as in NCB on CLEVR-Hans), we use permutation-invariant Set Transformers for Merlin, Morgana, and Arthur. For fixed-size concept vectors (as in CLIP/SpLiCE on CIFAR-100, ImageNet-1k, and COCOLogic-10), we use shallow nonlinear MLPs.

In preliminary experiments, we observed that performance is stable across a range of nonlinear architectures (varying depth, width, and activations), whereas purely linear models consistently underperformed by collapsing to simple correlation tests. Based on this, we recommend using permutation-invariant architectures for slot-based concepts and nonlinear MLPs for vector-based concepts when applying NCV to new datasets.

### E.4   EFFECT OF THE WEIGHTING PARAMETER $\gamma$

The weighting parameter $\gamma$ controls the trade-off between Merlin's cooperative objective and Morgana's adversarial objective when training Arthur. Recall that Arthur's loss is given by

$$L_A = (1 - \gamma) L_M + \gamma L_{\widehat{M}},$$

so that $\gamma = 0$ corresponds to training Arthur only on Merlin's loss, and larger $\gamma$ increases the relative weight of Morgana's adversarial objective. To assess the influence of this trade-off on NCV, we sweep $\gamma$ over a range from 0 (Merlin-only) up to 0.5 (substantial weight on Morgana) on CIFAR-100, ImageNet-1k, and COCOLogic, keeping all other hyperparameters fixed. For each setting, we train three models with different random seeds and report the mean completeness and soundness; for COCOLogic, we report balanced metrics due to class imbalance.

Across all datasets the same qualitative behavior emerges. When $\gamma = 0$, i.e., the verifier is trained without adversarial pressure from Morgana, completeness remains high but soundness collapses (e.g., 37.9% on CIFAR-100, 10.4% on ImageNet-1k, and 51.8% balanced soundness on COCOLogic). As soon as $\gamma > 0$, soundness rapidly recovers to values close to those reported in Tab. 1, while completeness remains essentially unchanged throughout the range of $\gamma$ we consider. These results demonstrate that incorporating adversarial concept selection is crucial for learning verifiers that remain

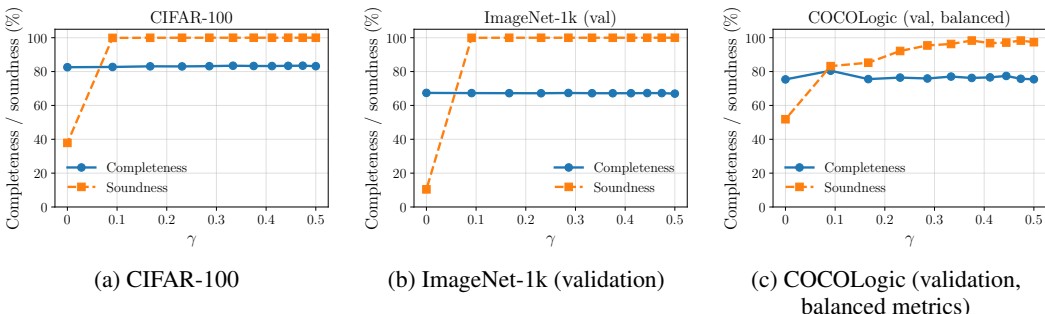

Figure 5: Effect of the weighting parameter $\gamma$ on completeness and soundness for (a) CIFAR-100, (b) ImageNet-1k validation, and (c) COCOLogic validation (balanced metrics). All curves show means over 3 random seeds.

reliable under misleading concept subsets, without sacrificing standard predictive performance in the regimes we study. For a detailed discussion of why an adversarial prover and its relative strength are essential for the Merlin–Arthur mutual-information guarantees underlying NCV, see Suppl. A.

# F  COMPUTATIONAL COST AND HARDWARE SETUP

To complement the main results, we report approximate training times for all models and datasets considered in Sec. 4. All runs were executed on a single NVIDIA A100 or A40 GPU; times are reported as rounded wall-clock estimates and are meant to convey relative cost rather than exact benchmarks. For CBM baselines, we *exclude* the per-sample SpLiCE optimization step at inference time, which would add substantial overhead and further increase their deployment cost.

Table 11: Approximate training time comparison across models and datasets. Times are rounded wall-clock estimates on a single NVIDIA A100 or A40 GPU. For CBM baselines, the per-sample SpLiCE optimization at inference time is omitted here, but would incur significant additional cost. (ImgNet = ImageNet-1k, COCO-10 = COCOLogic-10, CL-H3/CL-H7 = CLEVR-Hans3/7.)

| Model | CIFAR | ImgNet | COCO-10 | CL-H3 | CL-H7 |
|---|---|---|---|---|---|
| ResNet (baseline) | ∼25–30m | ∼1.5d | ∼50m | < 10m | < 10m |
| Pixel-MAC | ∼1.3d | ∼3d | ∼4h | < 2h | < 2h |
| CBM | ∼10–15m | ∼4h | ∼2m | ∼2m | ∼2m |
| **NCV (ours)** | ∼20m (11m A + 7m PVG) | ∼5h (2.5h A + 2.5h PVG) | ∼5m | ∼1.5m | ∼5m |

Overall, NCV is 1–3 orders of magnitude cheaper to train than Pixel-MAC across all datasets, while remaining comparable to standard CBMs in runtime. Since concept encodings are precomputed once and reused across models, NCV scales similarly to a conventional classifier without additional architectural overhead, and remains practical even in large-scale settings such as ImageNet-1k.

# G  USE OF LARGE LANGUAGE MODELS

Large language models were used to support this work by assisting with text refinement, implementation of code components (including methods and plot generation), and by providing input during idea development and approach refinement.

