# OpenReview forum: "Neural Concept Verifier: Scaling Prover-Verifier Games via Concept Encodings"
_ICLR.cc/2026/Conference — Submitted to ICLR 2026_

### Official Review · Reviewer_2XZ1 · 2025-10-21

**Soundness:** 2
**Presentation:** 3
**Contribution:** 2
**Rating:** 2
**Confidence:** 4

**Summary:**

Concept-based models (CBMs) combine neural concept extractors with a classifier that only takes concepts as input. The latter classifier is often a simple linear output layer, limiting expressivity and reducing the accuracy compared to end-to-end neural network models. This paper proposes to replace that linear layer with a "Merlin-Arthur classifier (MAC)", a recent GAN-like formalism where a neural network model selects certain concepts, and another neural network model uses that subset of features to classify the output. This MAC is claimed to retain interpretability.

**Strengths:**

- Novel approach, I did not know about these MACs.
- Good amount of clearly described experiments with evidence for the method.
- Writing is overall clear and easy to follow

**Weaknesses:**

- Fails to discuss and compare to other works that tackle the same limitation of CBMs
    - Because of this existing body of literature, somewhat incremental
- It is unclear what the theoretical benefits of interpretability and verifiability of MACs are
- A few vital details for reproducability are missing
- No code provided

**Questions:**

Critical:
- The authors claim "the classifier component of CBMs has received little attention" (107). However, this has already been studied since 2022 [1], including particularly the accuracy-interpretability tradeoff. Several papers followed up on this, using various classifiers that have different tradeoffs. Here are at least a few I'm aware of [2-4]. None of these are mentioned in the manuscript.
    - I believe there should at least be a baseline of some of these papers (at least [1, 3], I'd say)
    - Furthermore, the related work should discuss the benefits of NCV compared to these existing approaches
- The abstract, introduction and conclusion mention the '(formal) verifiability' of NCV by using a 'verifiable classifier'. However, the manuscript contains no clear theoretical claims as to what exactly can be verified. I guess these are inherited by using MACs, but it should be clearly stated in the paper with the relevant assumptions needed.
- The above criticism also holds for the claim that "NCV emphasizes faithfulness and interpretability" (253). Why are explanations faithful? And why are they interpretable? As far as I understand, there is nothing preventing the Arthur model from completely misunderstanding the concepts as long as it gets high classification accuracy.
- The loss is not differentiable with respect to Merlin and Morgana, as it involves discrete subset selection. I could not find how NCV estimates gradients for these models.

If the above comments are resolved, I would be happy to increase my score.

Other comments and questions
- The paper mentions a 'rejection class' to use for adversarial inputs. However, it is not clear why it would ever predict it, as it is not the case the model has to predict 'reject' for Morgana.
- Q3 about 'more detailed explanations' was surprising. NCV does not provide more detailed explanations to me, rather more abstract and higher-level than pixel level. Further, the claim that 'NCV ... pinpoints the precise attributes relevant for a class decision" (407-408) seems unproven from 5 qualitative examples, and it is unclear if these explanations are causally reliable [5] / causally opaque [4].

1. Espinosa Zarlenga, M., Barbiero, P., Ciravegna, G., Marra, G., Giannini, F., Diligenti, M., ... & Jamnik, M. (2022). Concept embedding models: Beyond the accuracy-explainability trade-off. Advances in neural information processing systems, 35, 21400-21413.
2. Debot, D., Barbiero, P., Giannini, F., Ciravegna, G., Diligenti, M., & Marra, G. (2024). Interpretable concept-based memory reasoning. Advances in Neural Information Processing Systems, 37, 19254-19287.
3. Barbiero, P., Ciravegna, G., Giannini, F., Zarlenga, M. E., Magister, L. C., Tonda, A., ... & Marra, G. (2023, July). Interpretable neural-symbolic concept reasoning. In International Conference on Machine Learning (pp. 1801-1825). PMLR.
4. Dominici, G., Barbiero, P., Zarlenga, M. E., Termine, A., Gjoreski, M., Marra, G., & Langheinrich, M. (2024). Causal concept graph models: Beyond causal opacity in deep learning. ICLR.
5. De Felice, G., Flores, A. C., De Santis, F., Santini, S., Schneider, J., Barbiero, P., & Termine, A. (2025). Causally reliable concept bottleneck models. arXiv preprint arXiv:2503.04363.

---

> ### Author Response · Authors · 2025-11-21
>
> # Response to Reviewer 2XZ1
>
> We thank the reviewer for their thoughtful and positive assessment of our work. We appreciate the recognition of the novelty of the approach, the clarity of the writing, and the strength of the experimental evidence. Below, we address each of the reviewer's concerns in detail and outline the corresponding revisions to the manuscript.
>
> ## Response to W1/Q1 -- Comparison with recent nonlinear and symbolic CBM extensions
>
> Thank you for raising this important point. We agree that our initial submission did not sufficiently contextualize NCV within the broader line of nonlinear, symbolic, and causal CBM extensions introduced since 2022, including Concept Embedding Models (CEMs) [1], Concept-based Memory Reasoning (CMR) [2], Interpretable Neural-Symbolic Concept Reasoning (DCR) [3], and causal concept models [4,5]. We have updated the related-work section accordingly; a structured overview is also included in our **General Response (G3)**.
>
> In brief, these works enhance the classifier component of CBMs by introducing nonlinear predictors [1], memory-based rule structures [2], symbolic reasoning engines [3], or causal constraints [4,5]. NCV differs conceptually: instead of designing a new predictor, NCV utilizes a prover–verifier training framework that enforces sparse, per-sample concept selection and evaluates predictions under competing subsets proposed by Merlin and Morgana. This mechanism is orthogonal to the cited approaches and is specifically designed to scale to large, automatically discovered concept spaces (1k–10k concepts), a setting that most of these models do not target.
>
> To directly address the reviewer's suggestion, we have implemented the DCR baseline [3] that also integrates Concept Embedding Models [1] with a proposed nonlinear classifier. This setup allows jointly learned concept representations together with a nonlinear—and still interpretable—predictor. Our results in **Table A (also seen in General Response G3)** show that, on large concept vocabularies such as CLIP-Sim, these baselines performances remain far from NCV on ImageNet-1k, CIFAR-100, and COCOLogic.
>
> **Table A. Direct comparison between DCR and NCV at 10k vocabulary size. NCV metrics correspond to completeness (accuracy).**
>
> | Dataset | DCR (10k vocab) | NCV |
> |---------|-----------------|-----|
> | CIFAR-100 | 51.76% | **83.32%** |
> | ImageNet-1k | n/a | **67.04%** |
> | COCOLogic-10 | 38.61% | **75.42%** |
>
> Our intuition is that DCR performs well on smaller settings but does not scale to the large, automatically discovered concept vocabularies we target, whereas NCV remains stable and efficient across all datasets. To ensure a fair comparison, we additionally performed a hyperparameter sweep (including smaller vocabulary sizes), and we will provide full experimental settings and training details in the revised manuscript and anonymized code release.
>
> Overall, we emphasize that NCV is not intended to replace these approaches. Rather, it provides a complementary mechanism that can, in principle, be applied on top of any concept representation or predictor. We thank the reviewer for pointing us to these relevant works, which have helped us position NCV more clearly within the recent literature, and we hope that the additional discussion and experiments sufficiently address the reviewer's concerns.
>
> ---
>
> ## Response to W2/Q2 -- Theoretical guarantees
>
> The theoretical guarantees of NCV are derived from the Merlin-Arthur framework (Waldchen et al., 2024). In binary classification, the existence of a winning strategy for Merlin and Arthur implies a lower bound on the mutual information between the selected concepts and the true label, formally guaranteeing that the decision relies on task-relevant information. While the multi-class extension relies on empirical "soundness" (robustness to the trained adversary Morgana), this proxy serves as a practical certification of robustness. We will clarify these inherited guarantees and their assumptions in Section 3 and Appendix. For more details please refer to our **General Response (G2)**.
>
> ---

---

> > ### Author Response · Authors · 2025-11-21
> >
> > ## Response to W3/W4 -- Reproducibility details and Code availability
> >
> > Full reproducibility is important to us, and the revised manuscript will make all details easily accessible.
> >
> > Appendix D and Appendix E already provide extensive descriptions of our experimental setup, including architectures for all agents (Set Transformer and MLP variants), training procedures, hyperparameters, loss weights γ, datasets, and implementation choices for both CLIP- and NCB-based concept extractors. We will make the references to these appendices more explicit in the main text.
> >
> > To complement this, we are finalizing an anonymized code repository containing the full NCV implementation, including training scripts, evaluation code, and configuration files for all datasets and baselines (including DCR). A link to the anonymized repository will be added to the revised manuscript in the coming days to ensure full reproducibility.
> >
> > We hope that these additions fully address the reviewer's concerns regarding reproducibility and make it straightforward for others to verify and build upon our results.
> >
> > ---
> >
> > ## Response to Q3 -- Faithfulness via Adversarial Training
> >
> > Thank you for raising this concern. We agree that, without additional structure, a nonlinear classifier could in principle misinterpret concepts while still achieving high accuracy. In NCV, faithfulness does not come from the classifier alone but from the interaction between Merlin, Morgana, and Arthur as detailed in our **General Response (G2)**.
> >
> > In short, the key mechanism is Morgana's adversarial role. Concept extractors often return both meaningful and spurious co-occurring concepts (e.g., an image of a boat often comes with sea). Without an adversarial prover, Merlin and Arthur can "cheat" by jointly relying on spurious concepts—this degeneration is well documented in prior work (Chang et al., 2019; Wäldchen et al., 2024).
> >
> > Morgana prevents this failure mode. She deliberately selects misleading yet plausible subsets (e.g., sea without boat) to test whether Arthur's prediction changes. If Arthur incorrectly relies on the spurious concept (sea), Morgana immediately exposes this by breaking completeness or soundness. During training, Arthur is therefore forced to base predictions on concepts that remain informative even under adversarial settings.
> >
> > In practice, this dynamic ensures that Arthur cannot silently rely on shortcuts: any spurious or co-occurring concept that does not independently support the label is systematically challenged by Morgana and eliminated as a reliable signal. The only concepts that remain stable across both provers' selections are those that are genuinely tied to the class. This is precisely why the resulting explanations are faithful: the sparse subsets that Merlin supplies (and Arthur relies on) are exactly those that withstand adversarial interventions. We will make this mechanism explicit in the revised manuscript. This interaction is the core of the PVG framework, and the theoretical guarantees for completeness, soundness, and their mutual-information interpretation build on this intuition.
> >
> > ---
> >
> > ## Response to Q4 -- Gradients for Discrete Selection
> >
> > While the concept selection is discrete, both Merlin and Morgana are trained using continuous mask logits, making their selection mechanism differentiable. During training, the provers first output soft masks, which are applied to the concept vector so that gradients from Arthur's loss can flow back into the provers. After the prover updates, these soft masks are discretized using top-k selection and the resulting hard subsets are passed to Arthur, who is then updated with respect to both Merlin's helpful and Morgana's misleading selections. This alternating scheme (soft masks for gradient flow and hard masks for Arthur's update) provides a stable and practical way to train NCV. We will add this implementation detail to the manuscript, and we thank the reviewer for bringing this point to our attention.
> >
> > ---
> >
> > ## Response to Q5 -- Rejection class
> >
> > The rejection class is used solely by Arthur, not by Morgana. When Morgana provides a misleading concept subset, Arthur may no longer have enough valid evidence to make a reliable prediction. In such cases, soundness requires Arthur to abstain rather than guess, and the rejection class is the mechanism through which Arthur signals that "this subset does not support any class with confidence."
> >
> > Likewise, if Merlin supplies a concept mask that does not meaningfully support the true class, Arthur is expected to abstain for the same reason. In practice, rejecting Merlin's subset lowers completeness, while rejecting Morgana's subset increases soundness. We will clarify this explicitly in the revised manuscript.
> >
> > ---

---

> > > ### Author Response · Authors · 2025-11-21
> > >
> > > ## Response to Q6 -- Detail of explanations
> > >
> > > Thank you for pointing this out. NCV indeed produces higher-level, sparse concept explanations rather than detailed visual descriptions, and we will adjust the wording accordingly. Merlin selects only a small number of semantically meaningful concepts, so the resulting explanations reflect these abstract units.
> > >
> > > The statement that NCV "pinpoints the precise attributes relevant for a class decision" refers to the fact that NCV identifies the concepts that remain reliably predictive of the class when the model is challenged with alternative concept subsets during training. This captures stability under the prover–verifier interaction, not causal attribution. NCV does not make causal claims and is not intended to address causal opacity as studied in [4, 5]. We will revise the manuscript to clearly separate NCV's soundness-based explanations from causal interpretability.
> > >
> > > ---
> > >
> > > ## Final Remark
> > >
> > > We thank the reviewer once again for their thoughtful feedback, especially the rich perspective on recent extensions of CBMs. We hope that the revisions outlined above address all concerns, and we will update the manuscript accordingly in the coming days. Once the revised version is uploaded, we will explicitly notify the reviewers so that the changes are easy to track.

---

> > > > ### Comment · Reviewer_2XZ1 · 2025-11-21
> > > >
> > > > I thank the authors for their extensive and clear rebuttal, which resolves most of my concerns. I look forward to the revised manuscript. A few clarification questions, which have a common theme :-)
> > > > - W2/Q2: This sounds nice, but I would have liked something a bit more formal and in the context of context-based modeling.  Eg what do these bounds on mutual information look like? I understand the goal of the paper is not to make new theoretical contributions, but since it's part of the claims of the paper I think it's good to make explicit what (existing) guarantees are and under what conditions, and most importantly if they are realistic in the CBM setting.
> > > > -  Same goes for Q3. I get the intuitions posed here, but without a formal statement on these guarantees it's a bit hard to be entirely convinced.
> > > > -  Q5: The intuition is clear, but I don't understand how the rejection class is used then during training (mathematically/ algorithmically)
> > > >
> > > > I'm happy to wait for the revised manuscript for updates on these questions!

---

> > > > > ### Author Response · Authors · 2025-11-27
> > > > >
> > > > > # Summary of revisions for Reviewer 2XZ1.
> > > > >
> > > > > We thank the reviewer for the thoughtful follow-up questions and for engaging so constructively with our rebuttal. The revised manuscript incorporates all requested clarifications and strengthens the theoretical and methodological exposition accordingly. We have uploaded the revised manuscript and highlighted all changes in orange for full transparency; only the newly added Appendix A remains unhighlighted to preserve readability.
> > > > >
> > > > > ---
> > > > >
> > > > > ## (W2/Q2) Formal guarantees and their scope in NCV settings
> > > > >
> > > > > Section 3.4 now references Appendix A.1–A.5 which now give the formal versions of the Merlin–Arthur guarantees we rely on, including: (i) the original binary-case mutual-information lower bound linked to completeness and soundness (Appendix A.1), (ii) the assumptions required for our multi-class instantiation (Appendix A.2), and (iii) a discussion of when these assumptions are realistic in the context of high-dimensional problems (Appendix A.5). This makes explicit what is guaranteed, under which conditions, and what the guarantees do *not* claim in the NCV setting.
> > > > >
> > > > > ---
> > > > >
> > > > > ## (Q3) From intuition to formal statement
> > > > >
> > > > > We strengthened Section 3.4 and Appendix A.1 by adding precise definitions of completeness and soundness, explicitly stating the soundness event we rely on—namely that Arthur must output either the correct class or the rejection class under Morgana's misleading subset. The connection between high completeness/soundness and our *notion of faithfulness* (Appendix A.3) of the selected concepts is now stated formally.
> > > > >
> > > > > ---
> > > > >
> > > > > ## (Q5) Use of the rejection class during training
> > > > >
> > > > > Appendix A.4 now includes a concise mathematical description of how the rejection class enters NCV's loss functions *and* how it is optimized in practice. The update scheme states that Merlin and Morgana are trained using soft masks for gradient flow, followed by top-$m$ discretization for Arthur's update, making clear how the min–max game is implemented with discrete concept subsets.
> > > > >
> > > > > ---
> > > > >
> > > > > ## (W1/Q1) Context within recent CBM extensions
> > > > >
> > > > > Section 2 (Related Work) and Section 4 (Experimental Evaluations) now position NCV among recent nonlinear, symbolic, and causal CBM extensions (e.g., CEM, CMR, DCR, causal concept graphs, causally reliable CBMs). We additionally include a DCR baseline trained on the same 10k-concept vocabulary for CIFAR-100, ImageNet-1k, and COCOLogic (reported in the rebuttal and in Appendix C.4, Table 4).
> > > > >
> > > > > ---
> > > > >
> > > > > ## (W3/W4) Anonymized code and reproducibility
> > > > >
> > > > > We now provide anonymized code in the supplementary material on openreview.
> > > > >
> > > > > ---
> > > > >
> > > > > Once again, we sincerely thank the reviewer for the constructive engagement throughout the rebuttal phase; this substantially strengthened the final manuscript. We hope these revisions address your remaining concerns and are happy to provide further clarification on any remaining points.

---

### Official Review · Reviewer_ACh6 · 2025-10-28

**Soundness:** 2
**Presentation:** 3
**Contribution:** 2
**Rating:** 6
**Confidence:** 3

**Summary:**

The paper introduces Neural Concept Verifier (NCV), a method that combines Concept Bottleneck Models (CBMs) with Prover-Verifier Games (PVGs) to improve robustness against shortcut learning while maintaining interpretability. NCV uses a concept encoder to map inputs to a concept space and employs a prover-verifier framework to ensure that predictions are based on meaningful concepts rather than spurious correlations. The method is evaluated on several image benchmarks, demonstrating improved robustness and interpretability compared to some existing approaches.

**Strengths:**

1. Framing prediction as a PVG on a concept bottleneck is an interesting approach to the problem of shortcut learning.
2. PVGs offer a different and more stable spin on adversarial training by shifting the adversarial intervention to the concept space instead of the input space.
3. The proposed method provides good semantic explanations without compromising predictive performance, which is a key challenge in interpretable ML.
4. The paper is easy to follow.

**Weaknesses:**

1. The claim that CBMs are "limited by their reliance on low-capacity linear predictors" is not quite fair; CBMs can easily be combined with non-linear predictors, so this is not really a serious limitation that this paper resolves.
2. The first 3 benchmarks in Table 1 are missing a key baseline, namely, CBM on the CLIP-Sim feature space with a nonlinear probe. This is to test whether the PVG approach has a distinct performance edge over simply applying CBMs on a rich concept vocabulary with nonlinear probes.
3. Table 2 is missing the Pixel-MAC baseline, which is important to see how NCV compares to other PVG methods on shortcut learning.
4. It is unclear how significant the robustness gains are since the paper assumes a safe (non-adversarial) concept encoder, which ignores an important risk/attack vector of shortcut learning where the concept encoder itself may learn spurious correlations.
5. (Minor) The last sentence of the Table 1 caption, "NCV consistently matches or outperforms baselines in completeness," is not totally accurate since the CBM baseline outperforms NCV on ImageNet-1k in terms of completeness.

**Questions:**

1. Can you provide an ablation study on the concept mask size? You report the results for 32 concepts, but it would be interesting to see the relationship between performance and the concept count.
2. Why do you think Pixel-MAC performs poorly on CIFAR-100, ImageNet-1k, and COCOLogic compared to CLEVR-Hans?
3. How was the CLIP-Sim vocabulary chosen? Was it selected based on relevance to the datasets, or was it a generic set of concepts?
4. Can you report the validation–test gap for Pixel-MAC?
5. How does the performance change if you only optimize Merlin's loss (i.e., $\gamma=0$)? Can you report the performance and loss curves for different values of $\gamma$?
6. Can you evaluate your method against input-space adversarial attacks (e.g., FGSM) to see if the robustness gains in concept space translate to input space robustness?
7. What are some real-world image-based settings where adversarial provers would be relevant? Real-world settings for pixel-level attacks are fairly established, but I'm not aware of ones where a concept selector may be compromised in the image domain.

---

> ### Author Response · Authors · 2025-11-21
>
> # Response to Reviewer ACh6
>
> We thank the reviewer for their constructive feedback and for highlighting the contributions of NCV, including the value of formulating prediction as a prover–verifier game over concepts rather than pixels, its ability to produce semantic explanations without sacrificing performance, and the overall clarity of the manuscript. Below, we address each concern in detail and revise the paper accordingly.
>
> ## Response to W1 -- Characterization of CBMs
>
> We agree with the reviewer that CBMs are not restricted to linear predictors; they can indeed be combined with nonlinear classifiers, and we will revise the wording to avoid implying otherwise. Our intention was to describe the standard CBM formulation used in most baselines—typically employing linear probes for interpretability—rather than to claim that nonlinearity is fundamentally incompatible with CBMs.
>
> At the same time, simply applying a nonlinear classifier to the concept bottleneck does not by itself yield an interpretable model. Recent work has addressed this by extending CBMs with nonlinear *yet interpretable* classification modules (e.g., DCR (Barbiero et al., 2023) and follow-ups), which learn global decision rules over the concept space. These approaches are orthogonal to ours and do not include prover–verifier interactions. To ensure fairness, we now compare against such nonlinear CBM variants (see our **general response (G3)**). In short, our results show that NCV scales effectively to large concept vocabularies and high-dimensional datasets, while DCR saturates or fails to train in these settings.
>
> Finally, we emphasize that our primary aim is not to compete with CBMs but to make PVG-style classification scalable to high-dimensional and logically complex datasets such as ImageNet-1k, CIFAR-100, and COCOLogic-10. Independent of this goal, our experiments indicate that NCV provides a promising, complementary direction to CBMs by enabling concept-space prediction within a prover–verifier framework.
>
> **References:**
>
> Barbiero et al. (2023) – *Interpretable neural-symbolic concept reasoning*. ICML, 2023.
>
> ---
>
> ## Response to W2 -- Missing nonlinear CBM baseline
>
> We agree and have indeed experimented with a nonlinear CBM variant using the same SpLiCE-based sparse concept representations followed by an MLP classifier (CBM+MLP), which we previously reported only in the shortcut-learning analysis (Table 3/4). Following the reviewer's recommendation, we will now include this baseline in the main results (Table 1) across all datasets.
>
> Briefly, our experiments show that CBM+MLP improves accuracy over linear CBMs, as expected from a more expressive classifier. However, in this setup the MLP remains a black-box predictor over the selected concepts and does come with a mechanism out-of-the-box to reliably identify which concepts drive the prediction or whether the decision relies on a minimal, interpretable subset, unlike DCR or NCV. In turn, NCV integrates sparse concept selection into a prover–verifier game, explicitly testing Arthur's predictions under competing subsets from Merlin and Morgana and thereby offering more structured, per-sample insight into which concepts support or challenge a decision. We will add the CBM+MLP results and clarify this distinction in the revised manuscript.
>
> ---
>
> ## Response to W3/Q4 -- Pixel-MAC in Table 2
>
> Pixel-MAC was initially omitted from the shortcut learning table because our focus in Table 2 was on concept-based models, and Pixel-MAC operates directly in pixel space, making its role in shortcut-shift evaluations different from the concept-space methods listed there. For completeness and transparency, we will add the corresponding Pixel-MAC validation–test results to Table 2 in the revised manuscript.
>
> ---
>
> ## Response to W4 -- Scope of Robustness Analysis
>
> Thank you for raising this important point. We agree that vulnerabilities in the concept encoder are an important concern. If the encoder itself captures shortcut-driven or spurious patterns, then any downstream method—including NCV—will inevitably receive a biased concept space.
>
> Importantly, NCV does not attempt to debias the encoder. Mitigating shortcut learning at the feature-extraction stage (e.g., through debiased pretraining or alternative extractors) is orthogonal to what NCV addresses. NCV focuses on the classifier's behavior given a concept space, evaluating whether predictions remain stable when Merlin's helpful concept subset competes with adversarially selected subsets by Morgana. This encourages reliance on concepts that remain informative.
>
> ---
>
> ## Response to W5 -- Caption correction
>
> We thank the reviewer for catching this inaccuracy. We will update the Table 1 caption to state that NCV matches or outperforms baselines "in most settings," acknowledging the specific case of ImageNet-1k where CBM slightly leads in completeness.
>
> ---

---

> > ### Author Response · Authors · 2025-11-21
> >
> > ## Response to Q1 -- Mask Size Ablation
> >
> > The requested ablation is already part of the manuscript: Appendix D (Tables 5 and 6) reports results for CLEVR-Hans-3 and CLEVR-Hans-7 across mask sizes $m \\in \\{4, 6, 12\\}$, and Appendix E (Tables 8 and 9) provides the corresponding ablations for CIFAR-100 and ImageNet-1k with mask sizes $m \\in \\{8, 16, 32, 64\\}$. These ablations are already referenced in the main paper, but we will make this reference more visible in the revised version.
> >
> > Briefly, across all datasets, we observe that increasing the mask size generally improves both completeness and soundness, as Merlin and Morgana can select richer concept subsets. However, larger masks reduce interpretability, since the selected concept sets become less sparse. For this reason, we use intermediate value of $m=32$ for CLIP-based experiments in Table 1, and smaller masks such as $m=12$ for NCB.
> >
> > ---
> >
> > ## Response to Q2 -- Pixel-MAC Scaling Challenges
> >
> > Pixel-MAC performs poorly on CIFAR-100, ImageNet-1k, and COCOLogic because the optimization landscape for the provers in raw pixel space ($32\\times32\\times3$ or $224\\times224\\times3$) is extremely difficult to navigate. The search space for meaningful pixel masks is vast, and gradient signals for discrete pixel selection are noisy and unstable. This difficulty is further amplified by the large number of classes in these datasets, which introduces substantial variability in the pixel patterns the provers must learn to select or suppress. These scaling limitations were a primary motivation for developing NCV, which shifts the prover–verifier interaction into a structured concept space where optimization is far more tractable.
> >
> > ---
> >
> > ## Response to Q3 -- CLIP-Sim Vocabulary
> >
> > For CIFAR-100, ImageNet-1k, and COCOLogic, we follow the setup of SpLiCE (Bhalla et al., 2024) and use a fixed textual vocabulary derived from the LAION-400M (Schuhmann et al., 2021) caption corpus: specifically, the 10,000 most frequent one- and two-word phrases. This choice ensures broad semantic coverage while remaining independent of any particular dataset, and it allows for a fair comparison with SpLiCE-based baselines since both methods rely on the same underlying vocabulary. We hope this clarification fully addresses the reviewer's question.
> >
> > ---
> >
> > ## Response to Q5 -- Excluding Arthur's loss term
> >
> > Thank you for this question. To isolate the role of Arthur's optimization, we conducted this ablation on the CLEVR-Hans-3 setup, freezing the pretrained Arthur (Arthur lr = 0) and updating only Merlin (and optionally Morgana). With Arthur fixed, the two provers act independently because the Arthur's decision function does not change. This allows us to analyze Merlin-only and Morgana-only behavior simultaneously without interference. The final row shows the full NCV setup on the same dataset, where Arthur is trained jointly with both provers and $\\gamma > 0$ encourages soundness against Morgana's misleading concept selections.
> >
> > | Arthur lr | Merlin lr | Morgana lr | $\\gamma$ | Train Comp. | Train Sound. | Val Comp. | Val Sound. |
> > |-----------|-----------|------------|---|-------------|--------------|-----------|------------|
> > | 0 | $10^{-3}$ | $10^{-3}$ | 0 | $61.25 \\pm 3.31$ | $25.16 \\pm 17.13$ | $60.70 \\pm 4.07$ | $24.07 \\pm 19.71$ |
> > | $10^{-3}$ | $10^{-3}$ | $10^{-3}$ | 1 | $99.18 \\pm 0.42$ | $99.89 \\pm 0.12$ | $97.47 \\pm 1.71$ | $99.59 \\pm 0.54$ |
> >
> > **Summary:** With Arthur frozen (pretrained but not updated via the prover-verifier setup), completeness can improve through Merlin's updates, but soundness collapses because Arthur never learns a stable decision boundary under adversarially chosen concept subsets. Only joint optimization of Arthur, Merlin, and Morgana (bottom row) yields both high completeness and high soundness.
> >
> > ### Effect of the weighting parameter γ
> >
> > We understand the reviewer's request as asking how performance changes when varying the trade-off between Merlin's cooperative objective and Morgana's adversarial objective in Arthur's loss. To evaluate this, we conducted a sweep over $\\gamma \\in \\{0.0, 0.1, 0.2, \\ldots, 1.0\\}$ on all three real-world datasets (CIFAR-100, ImageNet-1k, and COCOLogic). Across all settings, we observe the same pattern: once $\\gamma > 0$, i.e., Morgana is included in the optimization, the system reliably converges to completeness and soundness values close to those reported in Table 1. In contrast, setting $\\gamma=0$ leads to high completeness but markedly reduced soundness on all datasets, demonstrating that adversarial pressure from Morgana is essential for learning a verifier that remains reliable under misleading concept selections. We will summarize these findings in the appendix.
> >
> > ---

---

> > > ### Author Response · Authors · 2025-11-21
> > >
> > > ## Response to Q6 -- Input-space Adversarial Attacks
> > >
> > > Thank you for the question. In NCV, soundness is defined entirely within the prover–verifier framework: Arthur must not commit to an incorrect prediction when presented with a misleading concept subset selected by Morgana. This notion captures stability under adversarial concept–subset selection and is distinct from robustness to *input-level* perturbations. A more detailed explanation of completeness, soundness, and their role in NCV is provided in our general response on verifiability (see G2).
> > >
> > > Because Merlin and Morgana provide only sparse subsets of the concepts active in a given data point, Arthur is encouraged to *abstain* whenever the selected subset does not contain enough information to support a reliable prediction. Soundness therefore measures whether Arthur can avoid being forced into a wrong label under such adversarially chosen subsets.
> > >
> > > Adversarial attacks such as FGSM operate directly on the pixels and can alter the concept extractor's outputs. Since NCV does not modify or adversarially train the concept extractor, any vulnerability of the encoder to input-space attacks is inherited. For this reason, soundness in NCV should not be interpreted as input-space robustness.
> > >
> > > We will make this distinction explicit in the paper and clarify that NCV evaluates *soundness* only with respect to adversarial *concept subset* selection. This notion is separate from input-level adversarial robustness and should not be interpreted as such.
> > >
> > > ---
> > >
> > > ## Response to Q7 -- Relevance of adversarial provers
> > >
> > > Thank you for the question. Situations where adversarial provers are relevant arise whenever concept extractors return co-occurring but misleading concepts. For example, in an image of a boat on the sea, both *boat* and *sea* may be activated, even though *sea* is not predictive of the class *boat*. Without Morgana, Merlin and Arthur can "cheat" by jointly relying on such shortcuts—this type of degeneration has been documented in prior work (Chang et al., 2019; Wäldchen et al., 2024).
> > >
> > > Morgana prevents this collapse by deliberately selecting misleading yet plausible subsets (e.g., sea without boat), forcing Arthur to base decisions on concepts that are actually class-informative. This mechanism becomes relevant in any real-world setting where concept extractors produce both meaningful and spurious concept activations—Morgana ensures that the learned classifier cannot silently rely on spurious concepts.
> > >
> > > We hope this clarifies the role of Morgana, and we are happy to provide further details in the revised manuscript if helpful.
> > >
> > > **References:**
> > >
> > > Wäldchen et al. (2024) – Interpretability Guarantees with Merlin–Arthur Classifiers. AISTATS 2024.
> > >
> > > Chang et al. (2019) – A Game Theoretic Approach to Class-wise Selective Rationalization. NeurIPS 2019.
> > >
> > > ---
> > >
> > > ## Final Remark
> > >
> > > We thank the reviewer once again for their thoughtful and constructive feedback. The comments helped us sharpen both the conceptual positioning of NCV and the clarity of its empirical evaluation. We will integrate all promised revisions into the updated manuscript over the coming days, including improved explanations, corrected terminology, expanded baselines, and clarified experimental details. We appreciate the reviewer's insights and hope that the revised version fully addresses all concerns; we remain happy to provide any additional information if needed.

---

> > > > ### Author Response · Authors · 2025-11-27
> > > >
> > > > # Summary of revisions for Reviewer ACh6.
> > > >
> > > > We have uploaded the revised manuscript and highlighted all changes in orange for full transparency; the newly added Appendix A is left unhighlighted to preserve readability. In the revised manuscript, we address your main concerns as follows:
> > > >
> > > > ---
> > > >
> > > > ## Characterization of CBMs and nonlinear classifiers
> > > >
> > > > The related work sections now explicitly acknowledge that CBMs can be combined with nonlinear predictors, and we rephrase our claims so that NCV is presented as a complementary prover–verifier framework rather than as removing a fundamental linearity limitation. In addition, Section 4 (Experimental Results) now position NCV among recent nonlinear, symbolic, and causal CBM extensions (e.g., CEM, CMR, DCR, causal concept graphs, causally reliable CBMs), and we include a DCR baseline trained on the same 10k-concept vocabulary for CIFAR-100, ImageNet-1k, and COCOLogic (see Appendix C.4, Table 4).
> > > >
> > > > ---
> > > >
> > > > ## Nonlinear CBM baseline (CBM+MLP)
> > > >
> > > > Additionally, we now include a CBM+MLP baseline in Table 1 that uses the same SpLiCE-based sparse concept representations followed by an MLP classifier, enabling a direct comparison between NCV and a more expressive but less transparent nonlinear CBM variant on CIFAR-100, ImageNet-1k, and COCOLogic.
> > > >
> > > > ---
> > > >
> > > > ## Verifiability, soundness, and scope of robustness
> > > >
> > > > Section 3.4 now introduces completeness and soundness as explicit probability metrics with displayed equations and clarifies that all claims of "verifiability" in the main text refer to these two quantities. Appendix A.1–A.3 further spell out the Merlin–Arthur guarantees and their assumptions (including the binary-case mutual-information bound), delimit the scope of these guarantees in our multi-class concept-based setting, and emphasize that our soundness notion captures robustness to Morgana's *concept-level* adversary under a fixed concept encoder, which is distinct from both pixel-level adversarial attacks and the shortcut-robustness metric used in the CLEVR-Hans experiments.
> > > >
> > > > ---
> > > >
> > > > ## Anonymized code and reproducibility
> > > >
> > > > We now provide anonymized code in the supplementary material on openreview.
> > > >
> > > > ---
> > > >
> > > > ## Additional experiments and clarifications
> > > >
> > > > - We make the existing mask-size ablations more visible from the main text pointing to corresponding supplementary material in Appendix D and E.
> > > > - We now clarify how the CLIP-Sim vocabulary is constructed in the main text and explicitly point the reader to Appendix E.1 for a more detailed explanation.
> > > > - We refine the Table 1 caption so that completeness claims are not overstated.
> > > > - We expand the appendix discussion on why adversarial provers are relevant (Appendix A).
> > > > - We add a new study (Fig. 5, App. E.4) showing the effect of the weighting parameter $\\gamma$ on completeness and soundness across all real-world datasets.
> > > >
> > > > ---
> > > >
> > > > We appreciate the reviewer's insightful questions and suggestions, which have strengthened our presentation. We hope these revisions fully address the raised concerns and welcome any further discussion.

---

> > > > > ### Comment · Reviewer_ACh6 · 2025-11-27
> > > > >
> > > > > I thank the authors for their thorough rebuttals and ablations. At the moment, I still have some major concerns:
> > > > >
> > > > > 1. The paper's measure of soundess as robustness to a specific adversay (Morgana) seems to be somewhat trivial. This is primarily due to the fact that Morgana is optimized to induce an Arthur misprediction on a **soft**-masked concept encoding (not the binary-masked one). This is arguably a huge mismatch with how Arthur is optimized since it operates on the binary-masked encodings. The soft masks give Morgana a lot of control over the concept activations since it's not just "selecting" N concept features, it's modifying the entire encoding vector. To my mind, that should make it much easier to minimize Morgana's *actual* loss while also mimizing Arthur's loss. There's no longer much tension between these two networks because the Morgana loss that Morgana actually optimizes is not the same the Morgana loss in Arthur's loss. And this is probably why all soundness scores across the paper are very high (close to 100%).
> > > > >
> > > > > 2. Figure 5 shows that the model's completeness score barely changes by increasing $\gamma$, even all the way up to $\gamma =1$ where Merlin's loss is completely absent. This is very bizarre to me. Do you have an explanation for that?
> > > > >
> > > > > 3. Tables 6 & 7 show very high (close to 100%) accuracies with a mask as small as 4 concepts. This suggests that most of the concepts are actually redundant in those benchmarks, so the robustness gains might just be a direct result of sparsity as opposed to any adversarial settings (Morgana in this case). If so, then the robustness gains are simply because of masking concepts.
> > > > >
> > > > > 4. There is not enough evidence that the robustness gains of incorporating Morgana transfer to other adversaries. I would suggest training a separate adversary from Morgana, say Marley, then train it to fool a trained-and-frozen Arthur and see if it stands the tests of robustness.

---

> > > > > > ### Author Response · Authors · 2025-11-28
> > > > > >
> > > > > > We thank the reviewer for engaging so carefully with both the original manuscript and our rebuttal, and for the thoughtful questions and sanity checks that substantially helped clarify and strengthen the paper.
> > > > > >
> > > > > > ---
> > > > > >
> > > > > > ## On soundness, soft masks, and the role of Morgana
> > > > > >
> > > > > > We appreciate the concern that soft-mask optimization could weaken the adversarial tension and make soundness appear trivial. The reviewer correctly noted that Arthur is always trained and evaluated on *hard* top-$m$ masks, while both Merlin and Morgana use soft masks only as a surrogate for the *selection* of concept subsets. This setup follows the optimization scheme already used by Wäldchen et al.; in this work, our focus is to investigate the benefits of moving the Merlin–Arthur prover–verifier interaction from pixel space into concept space.
> > > > > >
> > > > > > Empirically, this does not collapse the game. Merlin's soft-to-hard training leads to binarized masks that are consistently informative, allowing Arthur to match or surpass nonlinear CBMs in accuracy across all datasets we investigate. Symmetrically, Morgana learns adversarial concept subsets (under the same sparsity constraint) on which Arthur is trained to abstain or stay correct; when we remove Morgana's pressure (e.g., $\\gamma = 0$), completeness remains high but soundness drops sharply, indicating that the observed high soundness stems from adversarial training rather than from the soft-mask surrogate itself.
> > > > > >
> > > > > > During development we also tried Gumbel–Softmax to optimize binary masks end-to-end for Merlin and Morgana, but found the resulting subset selection noticeably less stable and overall worse than the scheme we adopt in this work.
> > > > > >
> > > > > > We thank the reviewer for this well-developed perspective on the experimental setup and agree that a more stable end-to-end discrete optimization method would further strengthen the framework and the optimization we perform; we now explicitly mention this as a promising direction for future work (Section 6, lines 521–524).
> > > > > >
> > > > > > ---
> > > > > >
> > > > > > ## Effect of $\\gamma$ and interpretation of Figure 5
> > > > > >
> > > > > > You are absolutely right to flag Figure 5 — in the first revised submission, the plot was **incorrect** due to a mismatch between the loss in Eq. (3) and the implementation used to generate the figure.
> > > > > >
> > > > > > In the newly revised submission, we **correct Figure 5** by aligning its horizontal axis with Eq. (3), while keeping the underlying runs unchanged. With this fix, completeness and soundness behave as expected.
> > > > > >
> > > > > > The corrected "Excluding Arthur's loss term" table is:
> > > > > >
> > > > > > | Arthur lr  | Merlin lr   | Morgana lr  | $\\gamma$ | Train Comp.          | Train Sound.          | Val Comp.            | Val Sound.           |
> > > > > > |------------|-------------|-------------|-----------:|----------------------|-----------------------|----------------------|----------------------|
> > > > > > | 0          | $10^{-3}$ | $10^{-3}$ | 0          | $61.25 \\pm 3.31$   | $25.16 \\pm 17.13$   | $60.70 \\pm 4.07$   | $24.07 \\pm 19.71$  |
> > > > > > | $10^{-3}$| $10^{-3}$ | $10^{-3}$ | 0.5        | $99.18 \\pm 0.42$   | $99.89 \\pm 0.12$    | $97.47 \\pm 1.71$   | $99.59 \\pm 0.54$   |
> > > > > >
> > > > > > These results confirm that adversarial pressure from Morgana ($\\gamma > 0$) is essential for achieving high soundness. We again thank the reviewer for their careful oversight in spotting this inconsistency.
> > > > > >
> > > > > > ---

---

> > > > > > > ### Author Response · Authors · 2025-11-28
> > > > > > >
> > > > > > > ## On sparsity, redundancy, and adversarial pressure
> > > > > > >
> > > > > > > We agree that the very high accuracies with masks as small as $m = 4$ in Tables 6 and 7 indicate redundancy in our concept vocabularies. However, this is intentional: in CLEVR-Hans-3/7, NCB discovers 64 unsupervised concepts per input, and Merlin/Morgana select subsets of size $m = 12$ (and $m \\in \\{4,6,12\\}$ in the ablation). With such an overcomplete vocabulary, it is expected—and desirable—that a small subset of concepts already supports high accuracy; our goal is precisely to explain predictions via compact, high-level concept sets, in line with the usual aims of concept-based models.
> > > > > > >
> > > > > > > Crucially, we do *not* attribute high soundness to sparsity alone. NCV is sparse for all values of $\\gamma$, but the $\\gamma$-sweep in Appendix E.4 shows a clear pattern: for $\\gamma = 0$, the model uses the same sparse masks and completeness remains high, yet soundness drops strongly (e.g., to $37.9\\%$ on CIFAR-100, $10.4\\%$ on ImageNet-1k, and $51.8\\%$ balanced soundness on COCOLogic). As soon as $\\gamma > 0$, soundness quickly returns to values close to those in Table 1, with completeness essentially unchanged. This indicates that high soundness comes from adversarial pressure via Morgana, not from sparsity alone.
> > > > > > >
> > > > > > > Intuitively, Morgana's role is to rule out "degenerate"[Chang et al., 2019] or "cheating" [Wäldchen et al., 2024]  solutions, where Merlin and Arthur rely on concept patterns that look reasonable under non-adversarial, cooperative masks but fail once an adversarial prover (i.e., Morgana) joins the game. Training Arthur against Morgana's adversarial subsets (rather than only optimizing Merlin and Arthur without Morgana) encourages behavior that remains reliable under deliberately misleading concept selections, which is exactly what our soundness notion is designed to capture.
> > > > > > >
> > > > > > > Finally, the theoretical guarantees in Appendix A.1–A.2 (instantiating the results of Wäldchen et al., 2024) explicitly require the presence of a reasonably strong adversarial prover. If we remove Morgana and only enforce sparsity, these guarantees no longer apply and the interpretability of the resulting explanations weakens. We acknowledge that the agents could, in principle, be made stronger (e.g., via more stable end-to-end optimization for discrete masks). Nevertheless, in our current implementation NCV already matches or surpasses strong interpretable baselines such as CBMs and nonlinear CBMs (e.g., DCR), while maintaining high soundness under adversarial concept selections.
> > > > > > >
> > > > > > > **References:**
> > > > > > >
> > > > > > > Wäldchen et al. (2024) – *Interpretability Guarantees with Merlin–Arthur Classifiers*. AISTATS 2024.
> > > > > > >
> > > > > > > Chang et al. (2019) – *A Game Theoretic Approach to Class-wise Selective Rationalization*. NeurIPS 2019.
> > > > > > >
> > > > > > > ---

---

> ### Author Response · Authors · 2025-11-28
>
> ## Transfer to other adversaries
>
> We appreciate the suggestion of evaluating NCV against an additional adversary ("Marley") that is trained post-hoc on a frozen Arthur. Conceptually, this is a natural extension, but it is also somewhat orthogonal to the specific guarantees we aim for in this work.
>
> First, our claims about "robustness" are deliberately scoped to the *soundness* notion defined in the paper:
>
> $$
>   \\text{Soundness}
>   \=\
>   \\mathbb{P}_{(x,y)\\sim\\mathcal{D}}\\!\\big[A(\\widehat{S})\\in\\{y,\\bot\\}\\big],
>   \\qquad
>   \\widehat{S} = \\widehat{M}(\\mathbf{c}) \\odot \\mathbf{c},
> $$
>
> i.e., Arthur avoids committing to a wrong label under *Morgana's* adversarial concept subsets, either by staying correct or abstaining. This is exactly the setting captured by the Merlin–Arthur theory we instantiate in Appendix A: the guarantees are phrased for one cooperative prover $M$ and one adversarial prover $\\widehat{M}$ of comparable strength. We do not claim robustness against arbitrary, stronger adversaries outside this threat model.
>
> Second, the proposed Marley experiment depends strongly on how Marley is defined. If Marley has essentially the same architecture, mask parameterization, and loss as Morgana and is trained post-hoc against a frozen Arthur, then Marley is effectively another instance from the same adversary class as $\\widehat{M}$. In this regime we expect a substantial degree of transfer by design: Arthur has been explicitly trained to be sound on *that* family of concept selections, so post-hoc retraining another similar adversary mainly tests optimization stability rather than a qualitatively new threat model.
>
> By contrast, if Marley is made substantially more powerful (e.g., a different architecture, a different perturbation space, or additional relaxations), then one is effectively moving beyond the Merlin–Arthur threat model: the theoretical framework no longer applies once the adversary class changes and the relative-strength assumptions are violated. As we clarify in Appendix A.1-A.5, our empirical soundness estimates are therefore guarantees *with respect to the trained adversarial prover class* (in line with the practical instantiation in Wäldchen et al., 2024), not against arbitrary worst-case attackers.
>
> ---
>
> ## Concluding remarks
>
> We again thank the reviewer for engaging so closely with both the manuscript and our rebuttal, for carefully sanity-checking our framework, and for spotting inconsistencies such as in Figure 5. We hope that the revised manuscript and the clarifications above address your concerns as precisely and transparently as possible. We would, of course, be happy to further discuss any remaining questions or issues that you consider important for the final assessment of our work.

---

### Official Review · Reviewer_xjBw · 2025-10-28

**Soundness:** 3
**Presentation:** 3
**Contribution:** 3
**Rating:** 6
**Confidence:** 4

**Summary:**

This work introduces **Neural Concept Verifier (NCV)**, a novel framework for interpretable image classification. The main contribution of this work is the integration of concept extractors into Prover-Verifier Game (PVG) frameworks achieving a joint training of three agents in the concept embedding space. Evaluations were done across 5 datasets, where different instantiations of NCV were tested (e.g. extractor=NCB, pgvs=Set Transformers). Results support the claimed strengths of the NCV framework, where SOTA accuracy is achieved on 4 out of 5 datasets with an additional certificate of robustness to shortcuts (low validation-test gaps). Output explanations are compared to pixel-MAC (an instantiation of PVG directly applied to image pixel space), showing human readable (text) concepts identified in the image that supports the predicted class.

In summary, this work reduces the interpretability-accuracy gap by demonstrating SOTA performance in image classification while outputting interpretable explanations.

**Strengths:**

S1: this work is well-motivated by the gap of sacrificing accuracy for interpretability or vice versa in current literature, and the results generally supported all the claims stated.

S2: the framework is designed to be generalizable -- it allows integration of different concept extractors and verifiers, showing the potential for future studies on this framework.

S3: scalability is studied on real-world image datasets such as ImageNet-1k.

S4: robustness for shortcuts is studied and discussed, where results support the claim of NCV being robust to shortcuts.

**Weaknesses:**

W1: sometimes it is not clear to the reader what is the difference between NCV and MAC as in section 3 it seems like most of NCV is just adapting MAC. The authors should clarify and emphasize on their innovations beyond MAC (for example, there are a couple places where the authors states ".... shifting the PVG to concept encodings ..." (line 343), it can be made more clear if the authors mention the original set up of PVG such as "... shifting the PVG from xxxxxx to concept encoding ..." to highlight the contribution.

W2: the reader assumes that the shown concepts/explanations were textual because the extractor's output was textual. This indeed shows interpretability, however, it is not clear to the reader whether the quality of these concepts were controlled. For example, in Figure 3 for the coffee-mug, "extend" and "support" were two concepts output before "cappuchino", which seems to be irrelevant in the image, defeating the purpose of being interpretable. It might help if sub-regions in the image could be linked to each concept, achieving a better explanation and especially trustworthiness.

W3:  it was not discussed how much NCV might cost more than the baselines, for example, training time, hardware requirements, set-ups needed for replication of this work.

W4: it is not clear to the reader how "Soundness" was evaluated in Table 1, is it the same as the Robustness as in Table 2? What was the metric used for soundness?

W5: on line 377 it was claimed that NCV improves over linear CBMs in both accuracy and robustness, however, in table 1 CBM was not even tested on robustness (results are shown as n/a).

**Questions:**

It will be very helpful if the authors could answer the following questions:

Q1: In figure 2, when the concepts were output as "xxxx and yyyy and zzzz", are the "and" used for the presence of the concept (totally relying on the extractor used)? What if sometime it is the absence of a certain feature that drives the prediction? How would NCV output explanations in those cases?

Q2: on line 167, is C (the number of discovered concepts) a fixed or learnable parameter?

Q3: How are provers and verifiers selected for different datasets/tasks? Suppose we would like to apply NCV to a new image dataset, is there any property or guideline on how one should select the model architectures for the verifier and prover combination? Any pilot studies or ablations done for different combination of prover+verifier for the same dataset? Ultimately, how much does the selection of the combination affect the performance of NCV?

Q4: see W3

Q5: see W4

---

> ### Author Response · Authors · 2025-11-21
>
> # Response to Reviewer xjBw
>
> We thank the reviewer for their encouraging and constructive feedback. We appreciate the positive assessment of our motivation, framework design, and experimental scope, and we address the raised concerns point by point below while updating the manuscript accordingly.
>
> ## Response to W1 -- Distinction between NCV and MAC
>
> We thank the reviewer for highlighting the need to clarify the distinction between NCV and the original Merlin-Arthur Classifier (MAC). As the reviewer has identified, the primary innovation of NCV is shifting the Prover-Verifier Game (PVG) from the high-dimensional, unstructured pixel space (as in Wäldchen et al. 2024) to a semantic, symbolic concept space. This shift is critical for two reasons: (i) it enables scaling PVGs to complex real-world image datasets (e.g., ImageNet-1k) where pixel-based MAC fails to optimize effectively, and (ii) it ensures that the resulting explanations are grounded in human-interpretable concepts rather than diffuse pixel masks. We will revise Section 3 throughout to explicitly state these contributions.
>
> ---
>
> ## Response to W2 -- Concept Quality and Grounding
>
> We acknowledge that the quality of NCV's explanations is intrinsically tied to the underlying concept extractor, e.g., their ability to localise regions or the quality of their concept vocabulary. We have mentioned this in the discussion section (ll. 462-468). Thus, for ImageNet-1k, we used the same fixed vocabulary (LAION-based) as for the CBM baselines to ensure a fair comparison, and this vocabulary occasionally includes noisy or overly generic terms (e.g., "extend"). Importantly, NCV itself does not assume textual concepts: it is compatible with any concept space. As shown in our CLEVR-Hans experiments, NCV can operate on object-centric slots that are spatially grounded. In this setting, the sparse concept selection directly identifies the relevant object-slot, which allows us to decode and visualize the chosen object, as illustrated in Figure 3.
>
> ---
>
> ## Response to W3/Q4 on Computational Cost
>
> We agree with the reviewer that reporting computational cost is important. In the revised manuscript, we will add a dedicated table in the appendix summarizing approximate training times for all methods and datasets (with exact mean and standard deviation values available). All experiments were run on NVIDIA A100 or A40 GPUs. For readability in the response, we provide the approximated runtimes below.
>
> **Approximate training time comparison across models and datasets. For CBM baselines, the per-sample SpLiCE optimization step is omitted here, but would add significant inference overhead.**
>
> | Model | CIFAR-100 | ImageNet-1k | COCOLogic-10 | CLEVR-H3 | CLEVR-H7 |
> |-------|-----------|-------------|--------------|----------|----------|
> | ResNet (baseline) | ~25-30 min | ~1.5 d if trained from scratch | ~50 min | <10 min | <10 min |
> | Pixel-MAC | ~1.3 d | ~3 d | ~4 h | <2 h | <2 h |
> | CBM | ~10-15 min | ~4 h | ~2 min | ~2 min | ~2 min |
> | **NCV (ours)** | ~20 min (11 min Arthur + 7 min PVG) | ~5 h (2.5 h Arthur + 2.5 h PVG) | ~5 min | ~1.5 min | ~5 min |
>
> **Conclusion:** NCV is 1–3 orders of magnitude cheaper to train than Pixel-MAC, while remaining comparable to CBM baselines in runtime. It scales similarly to a standard classifier without introducing architectural overhead. Concept inference is a one-time amortized step and is not part of per-model training, which makes NCV practical even in large-scale settings.
>
> ---
>
> ## Response to W4/Q5 -- Soundness Evaluation
>
> We appreciate the opportunity to clarify our metrics. For a compact overview of how completeness, soundness, and verifiability relate in NCV, we refer the reviewer to our **general response provided in (G2)**. Here, we summarize the essential points.
>
> In NCV, each prediction is evaluated through a game between two provers over concept subsets from the *same input*: Merlin selects a helpful subset, while Morgana selects an equally sparse misleading one. In Table 1, *Soundness* measures the probability that the classifier (Arthur) does *not commit to an incorrect label* when presented with Morgana's misleading subset. In such cases, Arthur must either still output the correct class or abstain (reject), rather than produce a high-confidence wrong prediction.
>
> In contrast, the metric reported in Table 2 captures generalization under shortcut-learning shifts and is unrelated to the prover–verifier game. These metrics therefore quantify different phenomena: soundness measures resistance to wrong predictions caused by misleading feature selection, whereas Table 2 reports stability under shortcut-based distribution shifts. To avoid confusion, we will rename the Table 2 metric (e.g., "Shortcut Stability" or "Shortcut Robustness") and formally define the soundness metric in Section 3.4 while clarifying the distinction in the experimental section.
>
> ---

---

> > ### Author Response · Authors · 2025-11-21
> >
> > ## Response to W5 -- Robustness Terminology
> >
> > We thank the reviewer for spotting this mistake. The sentence at line 377 incorrectly suggested that NCV improves over linear CBMs in terms of robustness/soundness, even though standard CBMs were not evaluated with this metric (marked "n/a" in Table 1). This wording unintentionally anticipated the shortcut results later reported in Table 2, but that conclusion does not belong in this context and will therefore be removed.
> >
> > ---
> >
> > ## Response to Q1 -- Conjunctions and Feature Absence
> >
> > The "and" shown in Figure 2 is not a logical operator learned by NCV. It simply lists the concepts that the extractor detected in the image and that Merlin chose to support the prediction. The wording therefore reflects Merlin's selected subset of available concepts, not a learned rule of the form "concept A AND concept B."
> >
> > NCV does not model negation explicitly. If the *absence* of a feature matters, this is handled by Merlin *not selecting* the corresponding concept. For example, if a class depends on "a cup without a handle," then the concept "handle" will never appear among Merlin's selected concepts—either because no handle is present or because selecting it would contradict the class. Morgana can only pick "handle" if it truly exists in the image; she cannot invent missing features. Arthur then learns that if such a conflicting concept appears, the sample should not be assigned that class. In this way, NCV handles absence through selective concept inclusion rather than explicit logical negation.
> >
> > We hope this clarification resolves the reviewer's question.
> >
> > ---
> >
> > ## Response to Q2 -- Definition of C
> >
> > The number of concepts $C$ is determined by the concept extractor and is fixed from the perspective of the NCV agents (Provers and Verifier). For the CLIP-based instantiation, $C$ is the size of the fixed vocabulary. For the unsupervised NCB instantiation, $C$ is the number of learned concepts (e.g., determined by the number of concept clusters in the pretrained NCB model). We will clarify in Section 3 that while $C$ can be a learned parameter of the concept extractor, it acts as a fixed input dimension for the subsequent NCV stage.
> >
> > ---
> >
> > ## Response to Q3 -- Choosing Prover and Verifier Architectures
> >
> > The choice of prover and verifier architectures is primarily guided by the structure of the concept representation. For unordered concept-slot encodings (as in NCB), we use permutation-invariant Set Transformers. For fixed-size concept vectors (as in CLIP), standard MLPs are sufficient because the dimensionality is much lower than in pixel-space.
> >
> > At the start of the project, we experimented with several alternative designs (linear models, shallow MLPs, low-rank layers, and different activation functions such as ReLU or GeLU). Across these variants, we observed that performance is generally stable as long as the prover and verifier are non-linear; linear models consistently underperformed by collapsing to simple correlation tests. Based on this, we recommend using permutation-invariant architectures for slot-based concepts and non-linear MLPs for vector-based concepts when applying NCV to new datasets.
> >
> > ---
> >
> > ## Final Remarks
> >
> > We thank the reviewer once again for the constructive and insightful feedback. The comments helped clarify key distinctions between NCV and prior work and improved several parts of the presentation, including the role of concept quality, the differences between NCV and MAC, how we report computational cost, and how we define and position soundness. All promised revisions will be incorporated in the updated manuscript over the coming days, and we are happy to provide any additional details the reviewer may find helpful as the revision progresses.

---

### Official Review · Reviewer_ahDo · 2025-11-01

**Soundness:** 2
**Presentation:** 3
**Contribution:** 2
**Rating:** 4
**Confidence:** 4

**Summary:**

This paper introduces the Neural Concept Verifier (NCV), a framework that integrates Prover–Verifier Games (PVGs) with Concept Bottleneck Models (CBMs) to achieve interpretable and verifiable classification on high-dimensional data such as images. By shifting the prover–verifier interaction from pixel space to concept encodings, NCV aims to combine the formal verifiability of PVGs with the interpretability of concept-based reasoning. The authors evaluate NCV on synthetic and real-world datasets (CLEVR-Hans, CIFAR-100, ImageNet-1k, and COCOLogic) and report improvements in completeness, soundness, and robustness to shortcut learning compared to standard CBMs and pixel-level PVG models.

**Strengths:**

The central idea, combining PVGs and CBMs, is intriguing

The focus on verifiability is timely and relevant for trustworthy AI.

The experimental setup is diverse, covering both synthetic and real-world datasets.

The exposition is generally clear and well organized, though sometimes too high-level.

**Weaknesses:**

**Lack of comparison with relevant recent models**
The paper compares only against plain CBMs and pixel-based PVGs. However, there has been a recent surge in models that incorporate interpretable yet nonlinear mappings—often grounded in logic or symbolic reasoning (e.g., [1], [2]). In particular, Debot et al. maintain a global logic-based task decoder that allows the use of theorem provers to provide formal proofs of desired logical properties. These works appear conceptually close to NCV, yet no theoretical and/or empirical comparison is provided.

[1] Barbiero, Pietro, et al. "Interpretable neural-symbolic concept reasoning." International Conference on Machine Learning. PMLR, 2023.

[2] Debot, David, et al. "Interpretable concept-based memory reasoning." Advances in Neural Information Processing Systems 37 (2024): 19254-19287.


**Unclear guarantees and metrics for verifiability**
The “soundness” metric is not formally defined. Since it depends on a jointly trained component, it is unclear what guarantees it offers against an external adversary. The Arthur verifier might be robust only to the attacks generated by Morgana, rather than to arbitrary ones. The underlying statistical assumptions should be explicitly stated.

**Ambiguous definitions of verifiability and robustness**
These are central to the paper’s claims, yet their formal definitions and relationship to completeness/soundness remain unclear. A deeper conceptual and formal grounding is needed.

**Unconvincing shortcut-learning setup**
The rationale for interpreting corruptions as shortcuts is unclear. The distinction between “clean samples” and “shortcut-free” data is not well defined. Conceptually, shortcuts arise when multiple predictive routes exist, making the human-desired one unidentifiable. This is akin to standard arguments in regularization theory. Adding corruption could instead eliminate or distort the “human” model entirely. The argumentation here is weak and conceptually confused. Moreover, adversarial setups can have a regularizing effect, which may confound the reported improvements. Comparisons with standard shortcut-mitigation strategies (e.g., regularization, reconstruction losses) would be more informative.

**Minor clarity issues**
- Figure 3 gives the impression that an object-centric model can segment individual objects in CLEVR-Hans, which does not seem to be the case. The same explanatory style as for ImageNet-1k would be preferable.
- The description of the shortcut experiments could benefit from a better description for clarity.

**Questions:**

- How does NCV theoretically or empirically differ from recent logic-based interpretable models (in particular with [2])?

- What formal guarantees does the “soundness” metric provide beyond internal consistency with the Morgana prover?

- Why should data corruptions correspond to shortcuts, and how does this align with the standard definition of shortcut learning?

---

> ### Author Response · Authors · 2025-11-21
>
> # Response to Reviewer ahDo
>
> We sincerely thank the reviewer for their thoughtful and constructive feedback. We appreciate the recognition of the central idea of combining prover–verifier games with concept bottlenecks, the relevance of focusing on verifiability for trustworthy AI, and the strengths of our experimental setup across both synthetic and real-world datasets. We also value the reviewer's appreciation of the clarity of exposition and their suggestion to further refine specific explanations. Below, we directly address each concern and revise the manuscript accordingly.
>
> ## Response to W1/Q1 -- Comparison with recent nonlinear / symbolic CBM extensions
>
> Thank you for pointing out these important recent developments. We agree that our initial submission did not sufficiently contextualize NCV with respect to nonlinear and symbolic concept-based methods such as Interpretable Neural-Symbolic Concept Reasoning (Barbiero et al., 2023) and Concept-Based Memory Reasoning (Debot et al., 2024). We have now expanded the related-work discussion and added a direct empirical comparison, as detailed in our **General Response (G3)**.
>
> These models enhance CBMs by introducing global rule-based or memory-based reasoning structures over concepts. NCV, in contrast, introduces a per-sample prover–verifier interaction that evaluates whether Arthur's prediction remains stable when Merlin's helpful concept subset competes with adversarial subsets from Morgana. This difference—global symbolic reasoning vs. per-example adversarial concept verification—makes the methods complementary rather than redundant.
>
> Following the reviewer's suggestion, we trained Barbiero et al. (2023) (which we refer to as DCR) on CIFAR-100, ImageNet-1k, and COCOLogic-10 using the same 10k-concept vocabulary as NCV, which is embedded using a Concept Embedding Model (CEM) (Espinosa Zarlenga et al., 2022). We additionally performed a hyperparameter sweep (e.g., smaller vocabulary sizes) to ensure a fair comparison, and we will provide full experimental details in the revised manuscript. This setup allows a direct comparison under identical concept representations. The results are shown below:
>
> **Table: Direct comparison between DCR and NCV at 10k vocabulary size. NCV metrics correspond to completeness (accuracy).**
>
> | Dataset | DCR | NCV |
> |---------|-----|-----|
> | CIFAR-100 | 51.76% | **83.32%** |
> | ImageNet-1k | n/a | **67.04%** |
> | COCOLogic-10 | 38.61% | **75.42%** |
>
> We observe that DCR performs well on smaller settings but does not scale to the large, automatically discovered concept vocabularies we target, whereas NCV remains stable and efficient across all datasets. We will include these results and a clarified comparison in the revised manuscript. All training and implementation details will be provided in the updated submission and in the anonymized code release.
>
> We thank the reviewer again for bringing this important line of work to our attention. Incorporating DCR as a baseline has strengthened the paper, and we appreciate the opportunity to position NCV more clearly within the broader landscape of nonlinear and symbolic concept-based reasoning.
>
> **References:**
>
> Barbiero et al. (2023) – *Interpretable neural-symbolic concept reasoning*. ICML, 2023.
>
> Debot et al. (2024) — *Interpretable Concept‐Based Memory Reasoning*. NeurIPS.
>
> Espinosa Zarlenga et al. (2022) - *Concept embedding models: Beyond the accuracy-explainability trade-off*. NeurIPS, 2022.
>
> ---

---

> > ### Author Response · Authors · 2025-11-21
> >
> > ## Response to W2/W3/Q2 -- Guarantees, Soundness, and Verifiability
> >
> > Thank you for raising this foundational point. A full clarification of verifiability, completeness, soundness, and the inherited guarantees from the Merlin–Arthur framework is provided in our **General Response (G2)**, which we will make explicitly accessible from the main paper. Here, we summarize the essential points.
> >
> > In NCV, soundness and completeness are defined exactly as in classical prover–verifier setups: Arthur must succeed under Merlin's helpful concept subset (completeness) and must not be forced into an incorrect prediction under Morgana's misleading concept subset (soundness). These metrics do not refer to pixel-level adversarial robustness; Morgana never perturbs the input, but adversarially selects alternative subsets of the concepts present in the same data point. In this sense, NCV evaluates whether the model identifies the concepts that remain reliably predictive of the class when the model is challenged with such alternative concept subsets during training, rather than robustness to arbitrary external adversaries.
> >
> > The theoretical connection to mutual information is inherited from MACs (Wäldchen et al., 2024): under specific assumptions, high completeness and soundness imply that the selected concepts carry substantial class-relevant information. We adopt the same assumptions—most importantly, that Merlin and Morgana have comparable expressive power, which we enforce via identical architectures. We will add formal definitions, cite the required assumptions, and clarify the scope of these guarantees in Section 3.4 and Appendix B.
> >
> > We hope this addresses the reviewer's concerns. All corresponding clarifications will be included in the revised manuscript, and we are happy to provide any additional details the reviewer may find helpful.
> >
> > ---
> >
> > ## Response to W3/Q3 -- Shortcut learning definition
> >
> > Thank you for the comment. We follow the standard notion of shortcut learning (Geirhos et al., 2020), where models rely on spurious but predictive features that work under the training distribution but fail once those correlations are removed. In CLEVR-Hans, the "corruptions" are not arbitrary perturbations but built-in shortcuts (e.g., object colors perfectly linked to a label). "Confounded" therefore denotes data where such a shortcut is present (e.g., cubes are always gray at train time, but random colors at test time), whereas "clean" or "non-confounded" denotes data without these confounding correlations (e.g., cubes are always gray both at train and test time).
> >
> > Moreover, NCV is tested across multiple "strengths" of shortcuts (Table 2), where we alter the number of "clean" samples seen at train time. In NCV the adversarial prover does not alter images; it simply selects alternative concept subsets from the same sample. This encourages the classifier to rely on features that remain informative even when shortcut-based features compete. Regularization-based mitigation suppresses spurious features, whereas NCV takes a different route by evaluating whether shortcut features can still mislead the prediction. We will clarify this terminology in the paper.
> >
> > **References:**
> >
> > Geirhos et al. (2020) - *Shortcut Learning in Deep Neural Networks*. Nature Machine Intelligence 2, 2020.
> >
> > ---
> >
> > ## Response to W4 -- Minor Clarity Issues
> >
> > Thank you for pointing this out.
> >
> > (1) **Figure 3:** In CLEVR-Hans we use object-centric concepts from the Neural Concept Binder (NCB). Because Merlin selects concepts sparsely, we can directly identify the object "slot" corresponding to the selected concept and decode it using NCB's pretrained decoder. This enables fully automated object-level visualization within NCV, without performing full semantic segmentation. In contrast, the ImageNet-1k examples operate at a semantic concept level based on CLIP embeddings and vocabulary concepts, rather than slot encodings, i.e., do not allow for specific object reconstructions. We will update the caption to clearly reflect this distinction and align the explanatory style.
> >
> > (2) **Shortcut experiments:** As promised in our response to W3/Q3, we will make the terminology ("clean'' vs. "confounded'') explicit and briefly clarify how the validation–test gap indicates shortcut reliance.
> >
> > ---
> >
> > ## Final Remarks
> >
> > We thank the reviewer once again for the thoughtful and constructive feedback. The comments helped us refine both the conceptual framing of NCV and its relationship to recent nonlinear and symbolic CBM extensions. We will incorporate all promised revisions into the updated manuscript over the coming days, including clearer definitions of completeness and soundness, a strengthened comparison to DCR and related models, and improved clarity around shortcut experiments and visualizations. If any further questions arise, we are happy to provide additional details or clarifications.

---

> > > ### Author Response · Authors · 2025-11-27
> > >
> > > ## Summary of revisions for Reviewer ahDo.
> > >
> > > We have uploaded the revised manuscript and highlighted all changes in orange for full transparency; the newly added Appendix A is left unhighlighted to preserve readability. In the revised manuscript, we address your main concerns as follows:
> > >
> > > ---
> > >
> > > ### Comparison to recent nonlinear / symbolic CBMs
> > > Section 2 (Related Work) and Section 4 (Experimental Evaluations) now position NCV among recent nonlinear, symbolic, and causal CBM extensions (e.g., CEM, CMR, DCR, causal concept graphs, causally reliable CBMs). We additionally include a DCR baseline trained on the same 10k-concept vocabulary for CIFAR-100, ImageNet-1k, and COCOLogic (reported in the rebuttal and in Appendix C.4, Table 4).
> > >
> > > ---
> > >
> > > ### Verifiability, completeness, and soundness
> > > Section 3.4 now introduces completeness and soundness as explicit probability metrics with displayed equations, and clarifies that all claims of "verifiability" in the main text refer to these two quantities. Appendix A.1–A.3 further spell out the Merlin–Arthur guarantees and their assumptions (including the binary-case mutual-information bound), delimit the scope of these guarantees in our multi-class setting, and emphasize that our soundness notion captures robustness to Morgana's concept-level adversary, which is distinct from both pixel-level adversarial attacks and the shortcut-robustness metric used in the CLEVR-Hans experiments.
> > >
> > > ---
> > >
> > > ### Shortcut-learning setup and metrics
> > >  The CLEVR-Hans / shortcut section in the experiments and the shortcut table caption have been rewritten to clarify what we mean by "confounded" vs. "clean" data, why the corruptions correspond to shortcuts in our setup, and how the validation–test gap should be interpreted; we explicitly explain Morgana's role in this context.
> > >
> > > ---
> > >
> > > ### Visualizations and minor clarity issues
> > > We added a clarifying footnote to Figure 3 that distinguishes NCB-based object reconstructions on CLEVR-Hans from CLIP-based high-level concept visualizations on ImageNet-1k.
> > >
> > > ---
> > >
> > > ### Anonymized code and reproducibility
> > >  We now provide anonymized code in the supplementary material on openreview.
> > >
> > > ---
> > >
> > > We thank the reviewer for their valuable feedback throughout the review process. We hope these revisions fully address all concerns and remain available for any further questions.

---

### Author Response · Authors · 2025-11-21
**General Response (Part A)**

# General Response (Part A)

We thank all reviewers for their valuable time and feedback. Before addressing each comment individually, we clarify several overarching points. An updated manuscript reflecting all promised revisions will be provided in the coming days.

## (G1) Summary Contributions

Neural Concept Verifier (NCV) builds on the prover–verifier framework, where the classifier is trained while being challenged by two provers selecting concept subsets for the same input. Merlin selects a sparse, helpful concept subset, and Morgana selects an equally sparse misleading subset; both are presented to the classifier Arthur. Arthur must maintain correct predictions under both selections. This interaction is evaluated using standard completeness and soundness criteria from prover–verifier frameworks (see (G2)).

A primary innovation of NCV is shifting the Prover–Verifier Game (PVG) from the high-dimensional, unstructured pixel space (as in Wäldchen et al. 2024) to a semantic, symbolic concept space. This shift is crucial because: (i) it scales PVGs to complex real-world image datasets (e.g., ImageNet-1k) where pixel-based MAC cannot be optimized effectively, and (ii) it ensures that resulting explanations are grounded in human-interpretable concepts rather than diffuse pixel masks.

## (G2) Soundness, Robustness, Verifiability, and Guarantees

In NCV, *verifiability* means that a prediction must be supported by a small, informative set of concepts, and these concepts must remain valid even when the model is challenged with alternative (potentially misleading) concept subsets during training. In practice, when the cooperative prover (Merlin) selects a helpful subset, the classifier should correctly predict the true label (*completeness*); and when the adversarial prover (Morgana) selects a misleading subset, the classifier must not be forced into a wrong label but either remain correct or abstain (*soundness*). Crucially, this notion of soundness does *not* involve perturbing the input, but purely the selection of a subset of existing concepts from the same sample.

Formally, let $c = g(x) \\in \\mathbb{R}^C$ denote the concept encoding of an input $x$, and let Merlin and Morgana produce sparse binary masks $m = M(c)$ and $\\tilde{m} = M_c(c)$, respectively. The verifier $A$ predicts only from the masked concepts $S = m \\odot c$ and $\\tilde{S} = \\tilde{m} \\odot c$, where $\\odot$ denotes element-wise masking. Completeness and soundness are defined as:

$$\text{Completeness} = \Pr_{(x,y)}\left[A(S)=y\right] \quad \text{and} \quad \text{Soundness} = \Pr_{(x,y)}\left[A(\tilde{S})\in\{y,\bot\}\right],$$

where $\\bot$ denotes rejection. Completeness evaluates whether the correct label can be justified by a sparse set of helpful concepts, while soundness ensures that no sparse, misleading concept subset can induce an incorrect decision.

At a high level, the underlying theory behind MACs (proven for the binary case in Wäldchen et al. 2024) shows that, under specific assumptions, high completeness and high soundness imply a lower bound on the *average mutual information* between the *selected features* and the *true class*, meaning that the selected concepts must carry substantial class-relevant information. One of these assumptions concerns the relative strength of the provers, which we support in NCV by giving Merlin and Morgana the same architecture and expressive capacity.

To avoid the ambiguity concerns which the reviewers have raised, we will (i) add formal definitions of completeness and soundness in Section 3.4, and (ii) include a related-work appendix section clarifying how prior prover–verifier theory yields mutual-information guarantees and what assumptions are typically required. Our goal is not to tighten these guarantees mathematically, but to make the prover-verifier framework practical at scale on high-dimensional and logically complex data (e.g., ImageNet-1k, CIFAR-100, COCOLogic, CLEVR-Hans). Finally, we aim to highlight that PVGs offer a promising, orthogonal direction to concept bottleneck models for interpretable classifiers, yet have received little attention due to past scalability limitations - limitations that NCV overcomes in practice while matching or outperforming CBMs in our experiments. For runtime comparisons, please see our response to reviewer xjBw (W3/Q4 - Computation Cost).

---

> ### Author Response · Authors · 2025-11-21
> **General Response**
>
> # (Part B)
>
> ## (G3) Additional Baselines & Related Work
>
> We sincerely thank the reviewers for pointing out important related works on nonlinear concept-based reasoning, such as (Espinosa Zarlenga et al., 2022), (Barbiero et al., 2023), and (Debot et al., 2024). We acknowledge that our discussion of the classifier component in CBMs was incomplete and have expanded our related work section to address this gap.
>
> Overall, we want to emphasize that NCV is orthogonal and complementary to the cited works rather than competing with them. The core contribution of NCV is not to propose a specific (non-linear) classifier for concept bottleneck models, but rather to introduce a framework that combines PVGs with concept-based representations to produce verifiable predictions supported by sparse concept evidence at scale. Crucially, NCV can be instantiated with any concept extractor and any predictor model, including the logic-based and neural-symbolic approaches the reviewers mention.
>
> To follow the reviewers' suggestion, we evaluated Interpretable Neural-Symbolic Concept Reasoning (Barbiero et al., 2023), which we refer to as DCR. DCR introduces a symbolic reasoning component on top of learned concepts. We trained DCR on CIFAR-100, ImageNet-1k, and COCOLogic-10 using the same 10k-concept vocabulary as NCV, embedded via a Concept Embedding Model (CEM) (Espinosa Zarlenga et al., 2022). This allows a direct comparison between rule-based reasoning and the prover–verifier mechanism of NCV.
>
> We report the results obtained with the 10k-concept vocabulary in Table 1. DCR successfully trains on CIFAR-100 and COCOLogic-10 but fails to complete training on ImageNet-1k—our intuition is that this is due to the combination of a large concept vocabulary and high-dimensional inputs, which leads to prohibitive training time and memory requirements. NCV, in contrast, scales robustly to all three datasets under the same vocabulary.
>
> **Table 1. Direct comparison between DCR and NCV at 10k vocabulary size. NCV metrics correspond to completeness (accuracy).**
>
>
> | Dataset | DCR | NCV |
> |---------|-----|-----|
> | CIFAR-100 | 51.76% | **83.32%** |
> | ImageNet-1k | n/a | **67.04%** |
> | COCOLogic-10 | 38.61% | **75.42%** |
>
>
> In summary, these results show that while logic-based reasoning models like DCR are effective on potentially smaller or curated concept sets, they appear to face scalability challenges when applied to large, automatically discovered vocabularies. NCV, in contrast, remains practical and effective in these settings by shifting the prover–verifier interaction into concept space and avoiding global rule-learning. As such, NCV complements rather than replaces nonlinear or symbolic CBM extensions.
>
> To ensure full reproducibility, we will include all training, implementation, and architectural details in the revised manuscript. In addition, we are finalizing an anonymized code repository containing the complete NCV implementation, including all baselines and experimental configurations. The anonymized link will be provided in the updated submission in the coming days.
>
> **References:**
>
> Barbiero et al. (2023) – *Interpretable neural-symbolic concept reasoning*. ICML, 2023.
>
> Debot et al. (2024) — *Interpretable Concept‐Based Memory Reasoning*. NeurIPS.
>
> Espinosa Zarlenga et al. (2022) - *Concept embedding models: Beyond the accuracy-explainability trade-off*. NeurIPS, 2022.

---

### Author Response · Authors · 2025-12-02
**Final Response**

## Final General Response to the Area Chair

We thank the Area Chair for taking over our submission under challenging circumstances and appreciate the effort invested in evaluating our work fairly. Below we provide a concise summary of how the revised manuscript addresses the core concerns raised in the reviews.


### 1. Clarified theoretical guarantees and prover–verifier structure
We substantially expanded Appendix A to give a precise and transparent account of how the Merlin–Arthur guarantees instantiate in our concept-based setting, including:
- how completeness and soundness are defined in concept space and related to the underlying information-theoretic guarantees (e.g., via asymmetric feature correlation \\(\\kappa_k\\) and mutual information bounds),
- how Merlin’s and Morgana’s masks, together with Arthur’s predictions over \\(\\{1,\\dots,K,\\perp\\}\\), induce the game and our notion of faithfulness of concept-based explanations,
- which assumptions from the original binary Merlin-Arthur Classifier (MAC) setting are preserved or relaxed in NCV’s multi-class, concept-level formulation, and which claims remain empirical, and
- the exact training losses for Merlin, Morgana (including the modified cross-entropy with rejection) and Arthur, and how the trade-off parameter \\(\\gamma\\) balances completeness and soundness.

These clarifications make the theoretical underpinnings and limitations of our guarantees explicit and directly connect the NCV formulation to the existing Merlin–Arthur theory.

### 2. Strengthened methodological clarity
The manuscript now includes:
- an improved explanation of why shifting Prover-Verifier Games (PVGs) into concept space enables optimization on high-dimensional datasets,
- an explicit description of how Morgana’s loss is stabilized and regularized,
- a clearer separation between concept extraction, sparse masking, and the nonlinear verifier, and
- a more structured step-by-step explanation of the NCV pipeline and improved notation.

These clarify the training dynamics and address earlier concerns regarding sparsity behavior and interpretability.

### 3. Expanded positioning within recent CBM extensions
We significantly improved the Related Work section and added a more complete comparison to nonlinear, symbolic, and causal CBM extensions. The revised manuscript:
- positions NCV as a complementary prover–verifier framework rather than as removing linearity constraints,
- includes a DCR (Barbiero et al., 2023) baseline trained on the same 10k-concept vocabulary for CIFAR-100, ImageNet-1k, and COCOLogic, and
- clarifies how NCV fits within the broader landscape of interpretable and verifiable learning.

### 4. Improved empirical evaluation and transparency
The evaluation section now provides:
- additional baselines,
- a clearer explanation of the \\(\\gamma\\)-sweep and its implications for completeness vs. soundness,
- ablations covering mask sizes and concept-vocabulary redundancy, and
- improved figure/table clarity and architecture descriptions.

The supplemental analysis requested by the reviewers has also been added.

### 5. Revised manuscript uploaded
All revisions described above are implemented in the updated manuscript. Changes are highlighted in orange for ease of inspection, while the new Appendix A is left unhighlighted to preserve readability.

---

### Closing remark
We are grateful for the detailed and constructive feedback on our submission. We believe the revised manuscript now offers a clear, rigorous, and comprehensive presentation of the *Neural Concept Verifier* and fully addresses the reviewers’ scientific concerns, and we trust that the revisions make these contributions and clarifications evident.


---

**Reference**

Barbiero et al. (2023) – Interpretable neural-symbolic concept reasoning. ICML, 2023.

---

### Meta-Review · Area_Chair_AVhf · 2026-01-13

**Summary:**

This paper introduces a framework that combines Prover-Verifier Games with Concept Bottleneck Models for interpretable image classification. The key contribution is shifting the prover-verifier interaction from pixel space to concept space, where a cooperative prover selects concept subsets that support classification while an adversarial prover selects misleading subsets, and a nonlinear verifier makes predictions based only on the selected concepts. The authors evaluate on synthetic benchmarks and real-world datasets, demonstrating competitive accuracy with concept-level explanations and some robustness to shortcut learning.

This paper received 4 reviews with mixed initial feedback (scores of 6, 6, 4, 2). On the one hand, reviewers appreciated the novel combination of prover-verifier games with concept-based models, the diverse experimental evaluation, and the writing. On the other hand, reviewers raised concerns about the practical significance of the verifiability claims, unclear theoretical guarantees, and missing comparisons with recent nonlinear CBM methods.

In this case, I am unfortunately recommending rejection. My decision is motivated by concerns about significance. The mathematical framework is sound, but its application feels contrived. What is missing is a compelling use case for the Prover-Verifier approach. As it stands, the paper primarily evaluates these models in terms of performance and provides a weak demonstrates that concept-level explanations for  shortcut mitigation.

I suspect that part of this stems from an unclear notion of what concepts are good for (this is a core issue within the literature on CBM). Maybe the best way to say this is that the "interpretability" we get from CBMs is a "means to an end." In an image classification task, for example, using a CBM would let us replace pixels with semantically meaningful concepts ("interpretability"). Given these concepts, humans can improve the model by checking that concepts are present at test time (e.g., concepts let us intervene). Another possible end is "alignment": given a CBM, we can easily control its behaves by enforcing constraints on its inputs or by adjusting their value (e.g., for steering).

I mention this because I fundamentally believe that the core idea of this work could be useful such use cases since it deals with how to select concepts. However, this would require a fundamentally different kind of framing and evaluation than the current paper. As it stands, the authors do not focus on the gains in accuracy from intervention or alignment (e.g., because the prover-verifier procedure selects concepts that support such tasks). In this case, I don't think that such a shift would be possible or guaranteed to succeed. It is also worth mentioning that these use cases are difficult to show off because the datasets used extract concepts from pretrained models (so the concepts do not correspond to ground truth concepts that would allow for perfect alignment or intervention).

**Reviewer Concerns:**

**Outstanding**
- Concept Quality Dependence [xjBw, 2XZ1]
- Theoretical Guarantees Scope [ahDo, 2XZ1]
- Soundness Interpretation [ahDo, 2XZ1, ACh6]

**Ignored / Resolved**
- Limited Baseline Comparison [ahDo, 2XZ1]

**Reviewer Scores:**

I don't see meaningful increases from from ahDo and 2XZ1. Potentially a 1 point increase in the other cases. Overall this should have remained a borderline deciison.

---

### Decision · Program_Chairs · 2026-01-26

Reject